# MAPS: Advancing Multi-Modal Reasoning in Expert-Level Physical Science

**Erle Zhu**[1,2], **Yadi Liu**[2], **Zhe Zhang**[2], **Xujun Li**[1,2], **Jin Zhou**[2],
**Xinjie Yu**[3], **Minlie Huang**[1,2], **Hongning Wang**[1,2,*]
[1]The Conversational AI (CoAI) Group, [2]Department of Computer Science & Technology
[3]Department of Electrical Engineering
Tsinghua University
zel24@mails.tsinghua.edu.cn, hw-ai@tsinghua.edu.cn

## Abstract

Pre-trained on extensive text and image corpora, current Multi-Modal Large Language Models (MLLM) have shown strong capabilities in general visual reasoning tasks. However, their performance is still lacking in physical domains that require understanding diagrams with complex physical structures and quantitative analysis based on multi-modal information. To address this, we develop a new framework, named **M**ulti-Modal Scientific Re**A**soning with **P**hysics Perception and **S**imulation (**MAPS**) based on an MLLM. MAPS decomposes expert-level multi-modal reasoning task into physical diagram understanding via a Physical Perception Model (PPM) and reasoning with physical knowledge via a simulator. The PPM module is obtained by fine-tuning a visual language model using carefully designed synthetic data with paired physical diagrams and corresponding simulation language descriptions. At the inference stage, MAPS integrates the simulation language description of the input diagram provided by PPM and results obtained through a Chain-of-Simulation process with MLLM to derive the underlying rationale and the final answer. Validated using our collected college-level circuit analysis problems, MAPS significantly improves reasoning accuracy of MLLM and outperforms all existing models. The results confirm MAPS offers a promising direction for enhancing multi-modal scientific reasoning ability of MLLMs. Our code is available at https://github.com/thu-coai/MAPS.

## 1 Introduction

Pre-trained on large-scale text and image corpora, Multi-Modal Large Language Models (MLLM) exhibit strong capabilities in general visual reasoning tasks, including image captioning and visual question-answering (Li et al., 2022; Team et al., 2023; AI; Liu et al., 2024). Through elaborated pre-training and post-training, the proficiency of LLMs in text-only mathematical reasoning and programming has significantly improved (Hendrycks et al., 2021; Lu et al., 2022; Lightman et al., 2023), broadening their applications to more scientific and professional tasks. However, for scientific disciplines that require understanding complex physical structures in images and mathematical reasoning based on scientific knowledge from multi-modal information, the capabilities of MLLMs remain weak (Yue et al., 2023). This limitation hinders their further application in educational, academic, and industrial scenarios. Thus, enhancing the multi-modal reasoning abilities of MLLMs in expert-level physical sciences while extending their application scenarios is a valuable yet challenging research direction.

The current methods in multi-modal reasoning (Zhang et al., 2023b; Zheng et al., 2023; Mitra et al., 2024) primarily concentrate on generating a rationale that integrates multi-modal information, allowing the model to derive the final answer from this intermediate result. This process is commonly referred to as Chain-of-Thought (**CoT**) (Wei et al., 2022) reasoning. Another commonly adopted pathway is to integrate LLMs with external tools, including small-sized specialized multi-modal models, as well as software such as code interpreter (Gao et al., 2023; Wang et al., 2024a). However,

---

*corresponding author

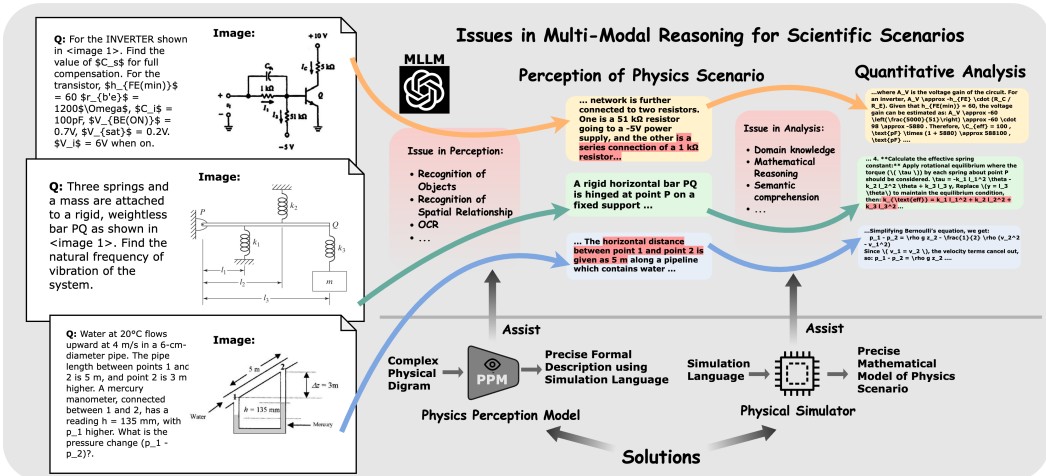

Figure 1: Two Issues in Multi-Modal Reasoning for Scientific Scenarios and Our Solutions. The location of the model error is highlighted in red. The scientific questions are sampled from MMMU (Yue et al., 2023).

these methods mainly focus on general images or diagrams containing simple physical information, making it difficult to directly transfer to scientific scenarios that involve complex physical diagrams and require precise numerical analysis.

To address the aforementioned limitations, we proposed **M**ulti-Modal Scientific Re**A**soning with **P**hysics Perception and **S**imulation (**MAPS**), a novel framework for solving complex multi-modal reasoning problems in physical disciplines. And in this paper we verified the effectiveness of MAPS in electrical discipline, which typically involves multiple circuit diagrams and is representative of expert-level physical science requiring reasoning on complex physical diagrams. The core idea of MAPS is to decompose expert-level reasoning problems into two sub-tasks: understanding the physical diagram and reasoning based on this comprehension and related physical knowledge. MAPS realizes physical diagram understanding by fine-tuning a visual language model using carefully designed synthetic data, resulting in what we term the Physics Perception Model (PPM). The role of PPM is to translate physical diagrams into simulation language descriptions that can be executed by a simulator. At the inference stage, MAPS integrates the converted simulation language description and their respective simulation results, obtained through a Chain-of-Simulation process, to derive intermediate rationales and ultimately the final answer to the question. Experiment results on college-level circuit analysis problems demonstrate that our framework can successfully address the challenges in complex multi-modal reasoning tasks in physical science. Most importantly, it significantly reduces the occurrence of hallucination when using and solving physical equations. This advancement creates new avenues for precise multi-modal scientific reasoning using MLLMs.

To summarize, our main contributions in this work are as follows:

- We introduce a novel multi-modal reasoning framework MAPS to address the current limitations of MLLMs in solving expert-level scientific problems involving complex physical diagrams. MAPS incorporates MLLMs with a finetuned perception model and physical simulator to improve the precision of its reasoning steps and results.

- Through our experiments on college-level circuit analysis problems, we demonstrate that MAPS significantly outperforms existing methods, offering a viable pathway to build multi-modal solutions for expert-level scientific problems.

- We devise an automated pipeline to synthesize diverse paired training data for finetuning an MLLM. By leveraging intrinsic generalization ability of pre-trained models, the pipeline helps MLLMs effectively adapts to complex real-world problems, alleviating the issue of data scarcity in scientific domains.

## 2   MOTIVATION

Following the human approach to solving science problems with diagrams, we break down the problem into two steps: understanding the physical context in the multi-modal input (**Perception**) and using scientific knowledge and mathematical deduction to derive the answer (**Analysis**). Based on these two steps, we summarize the limitations of current MLLM-based solutions for solving such problems into two main categories:

*Issues in Perception*. Based on observations reported in the MMMU benchmark (Yue et al., 2023) and our empirical studies, we found that current general purpose MLLMs, including the most powerful ones such as GPT-4V and Claude-3.5, exhibit poor perception abilities in understanding diagrams related to physical sciences (e.g., circuit diagrams). This corresponds to the perceptual error identified in the error analysis in (Yue et al., 2023). This significantly limits their application in the field of scientific reasoning with multi-modal input.

*Issues in Analysis*. Although MLLMs can sometimes correctly understand diagrams, their domain knowledge and mathematical reasoning abilities can still be lacking. This often leads to hallucinations during further reasoning steps, resulting in misleading answers.

We offer some specific cases in Figure 1, illustrating how an off-the-shelf MLLM makes mistakes in perception and analysis steps. Based on these observations, we decide to decompose this complex multi-modal reasoning task into sub-tasks and leverage expert models and domain-specific tools to solve the sub-tasks that are infeasible for current MLLMs. Concretely, as shown in Figure 1, our proposed two solutions to the two issues mentioned above are:

**Solution to *Issue in Perception*: Translate physical diagrams into simulation language descriptions.** We adopt simulation language for two reasons. First, it describes the physical scene of the diagram using a formal language, enabling the language model to directly access the fundamental structure behind the question. Second, with parameters of physical objects provided, we can directly use a simulator to obtain all states and observation values of the physical scene. In the context of circuit analysis, we use SPICE (Nagel, 1975) as our simulation language. For other scenarios, there are corresponding choices, such as ANSYS APDL (Kohnke, 1982) in mechanical disciplines and ZPL (Laikin, 2018) in optics domains. Specifically, we develop an expert visual language model to complete this conversion. Since there is no large-scale available dataset or existing model for this task, we devise a data synthesis pipeline to generate abundant physical diagrams and their corresponding simulation languages for our visual language model training.

**Solution to *Issue in Analysis*: Reasoning under the assistance of simulation.** Although current MLLMs can perform mathematical reasoning using external tools (Chen et al., 2022; Zhou et al., 2023), recent research found it still challenging to prompt LLMs to write programs for solving scientific problems (Tian et al., 2024). In the benchmark evaluating real-world scientific programming tasks, even the best model achieves an accuracy of less than 10% in completing a main problem. To address the issue of hallucination when MLLMs perform mathematical derivations and synthesize scientific programs, we delegate the main quantitative reasoning tasks to a domain-specific tool, namely a physical simulator. The simulator comprises domain-specific knowledge and thus is guaranteed to be precise in its output with respect to the given input.

Combining the two solutions above, we design a **Chain-of-Simulation** process that obtains simulation language description and simulation results utilizing the fine-tuned perception model and simulator, and prompt an MLLM to compute the answer under the assistance of simulation language description and simulation results at the inference stage.

## 3   METHODOLOGY

Our proposed MAPS framework, illustrated in Figure 2, consists of two phases: the **Physics Perception Model (PPM) construction phase** and the **Inference phase**.

The core components of our framework are as follows:

- **Physics Perception Model (PPM).** It serves as an expert perception model that translates a given physical diagram into a simulation language (SL) description. This model is fine-tuned from a pre-trained Visual Language Model (VLM) using a synthetic dataset designed for the diagram-to-SL conversion.

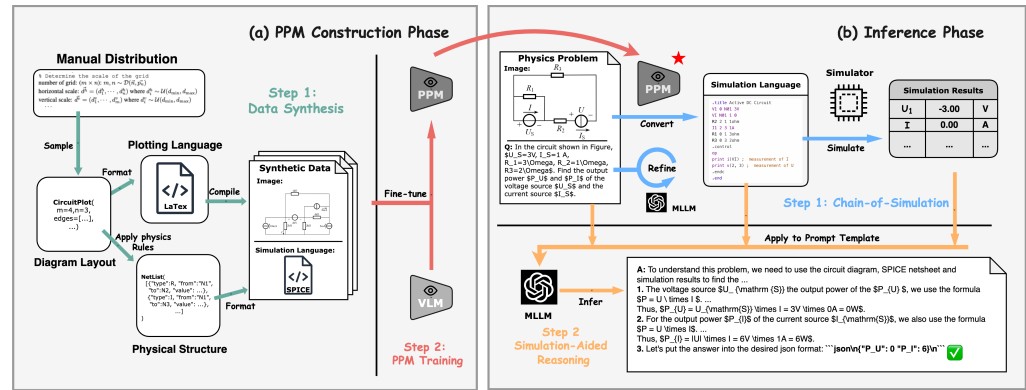

Figure 2: Our proposed **MAPS** framework is integrated with a Multi-modal Large Language Model (MLLM), a Physics Perception Model (PPM) and a Physical Simulator. **(a)** At **PPM Construction Phase**, we fine-tune a pre-trained VLM with carefully designed synthetic data to obtain PPM which can convert physical diagram into simulation language descriptions. **(b)** At **Inference Phase**, we apply **Chain-of-Simulation** to acquire simulation language description and simulation results which assist MLLM with the further reasoning to obtain final answer of original problem.

- **Physical Simulator.** The simulator is used to perform numerical simulations and obtain the state and observations about the physical scene carried in the diagram.
- **Multi-modal Large Language Model (MLLM)**. The MLLM primarily handles semantic understanding and basic mathematical reasoning, based on the results provided by PPM and the simulator. When solving physical problems with diagrams, the MLLM parses the target problem from the textual question, refines the simulation language description generated by the PPM, extracts useful simulation results, and performs final reasoning based on the original question and the added simulation information.

## 3.1 INFERENCE PHASE

We first introduce the inference phase because it conveys the main philosophy of our solution. As shown in Figure 2(b), this stage includes two steps: Chain-of-Simulation and Simulation-Aided Reasoning. Suppose we have a scientific problem with a physical diagram $X_V$ in a pixel format and textual description $X_L$, our model is required to infer the answer $Y_L$.

### 3.1.1 CHAIN-OF-SIMULATION

The first step in the Chain-of-Simulation (CoS) process is to use the PPM to convert the pixel schematic diagram $X_V$ into an initial SL description $Z$. Since the problem involves multi-modal information, the initial SL description produced by the PPM may lack completeness in depicting the full physical scene. For example, in a circuit diagram, a resistor might be labeled as $R_1$, but its value might be provided in the accompanying textual description in the question $X_L$. To address this, we employ the MLLM to refine initial SL description based on textual input $X_L$. The MLLM incorporates additional information from the accompanying text, resulting in a comprehensive and accurate SL text that fully describes the physical scene.

---

**Algorithm 1** MAPS: Inference Phase

1: **Input:** $X_V$, $X_L$, PPM, Simulator, MLLM
2: **Output:** $Y_L$
   % Chain-of-Simulation
3: Obtain SL description $Z \leftarrow \text{PPM}(X_V)$
4: Refine SL description
     $Z \leftarrow \text{MLLM}(X_L, Z, \texttt{prompt\_refine})$
5: Obtain simulation result $R \leftarrow \text{Simulator}(Z)$
   % Simulation-Aided Reasoning
6: **if** check\_valid($R$) **then**
7:    $Y_L \leftarrow \text{MLLM}(X_L, X_V, Z, R, \texttt{prompt\_sar})$
8: **else**
9:    $Y_L \leftarrow \text{MLLM}(X_L, X_V, Z, \texttt{prompt\_sl})$
10: **end if**
11: **return** $Y_L$

---

Once the comprehensive SL description $Z$ is generated, it is fed into the physical simulator to perform physical simulations. This process produces simulation result $R$, including state values and observation values of the physical scene. This approach effectively mitigates mathematical reason-

ing errors that may arise from the model's hallucinations in scientific computation, ensuring accurate and reliable results.

### 3.1.2 SIMULATION-AIDED REASONING

After the CoS process, MAPS will apply the question information $(X_L, X_V)$, SL description $Z$, and simulation result $R$ to a well-designed prompt template. This template prompts the MLLM to generate further rationale and infer the final answer $Y_L$. We consider the simulation language and simulation results as intermediate rationale in the model's reasoning process, similar to various Chain-of-Thought (CoT) mechanisms in existing prompting methods (Mitra et al., 2024; Zhang et al., 2023b; Zheng et al., 2023). The entire process is illustrated using pseudo-code in Algorithm 1. Experiments show that under the assistance of CoS, the MLLM can be prompted to accurately answer questions, effectively narrowing the gap between current MLLM capability and expert-level performance on scientific problems.

### 3.2 PPM CONSTRUCTION PHASE

Accurate conversion from pixel schematics to simulation language descriptions is crucial for the CoS to function effectively. We highlight this importance with a red star in Figure 2, emphasizing the significance of PPM in our framework. Due to the scarcity of real-world paired data that maps physical diagrams to simulation language descriptions, which is crucial for training Vision-Language Models (VLMs) to recognize the physical diagrams of interest, we choose to synthesize the paired data. To achieve this, we craft rules to generate a large dataset comprising diverse circuit diagrams and their corresponding simulation language descriptions. These synthetic data are then used to fine-tune the pre-trained VLM, ultimately producing the PPM.

### 3.2.1 DATA SYNTHESIS

The data synthesis process is depicted in Figure 2(a). And the detailed steps of a generation process are described as follows.

The diagram layout is our data structure designed to correspond to the plotting language, encompassing all the physical objects, their displayed positions and annotations in the diagram. Subsequently, the pipeline synthesizes the diagram and the corresponding SL description through two paths: the **diagram synthesis path** and the **simulation language (SL) synthesis path**.

**Diagram synthesis path**. As shown in the upper branch of Figure 2(a), the diagram layout is first converted to a plotting language. There are various plotting languages available, such as LaTeX (TikZ) and Graphviz, which use formal syntax to describe diagrams and can be compiled into pixel images. The design of diagram layout allows for a straightforward transformation from diagram layout to plotting language. Finally, we compile the generated plotting language using its designated compiler to generate the diagram in pixel format.

**SL synthesis path**. This path focuses on distilling the physical structure from the diagram layout using physical knowledge. Operationally, we apply physical rules to the diagram layouts to derive the intrinsic physical model, which contains only abstract physical objects and the functional relationships between them. For example, in circuit diagrams the physical structure can be formulated using a netlist model (Nagel, 1975), which includes all components along with their types, parameters, and topological connections. In mechanical scenarios, the physical structure can be described using a FEM (Rao, 2010) model to represent the mechanical system. Eventually, the physical structure is formatted into simulation language.

This process produces both a physical diagram and its corresponding simulation language description. Since each step of the generation procedure involves random sampling, a large number of diagram with different objects, spatial relationships and annotations can be generated through sufficient sampling.

### 3.2.2 PPM TRAINING

The training goal of the Physics Perception Model (PPM) is to generate the corresponding SL description from a given diagram. We use a decoder-only pre-trained visual language model as the base model for the PPM. In practice, the training loss during PPM fine-tuning is the average negative Maximum Likelihood Estimation (MLE) loss (Bishop & Nasrabadi, 2006) over the synthetic data.

## 4 EXPERIMENTS

We evaluate our MAPS framework through extensive experimentations on real-world scientific problems. Given the substantial workload involved in constructing and validating the entire pipeline, we have limited our initial verification to the circuit analysis scenario, which is generally believed very difficult for state-of-the-art MLLMs (Yue et al., 2023).

### 4.1 IMPLEMENTATION

In this section, we describe our implementation of MAPS framework in the circuit analysis scenario.

**Synthesis of Training Data for PPM**. In the context of circuit diagrams, the diagram layout is defined as the planar grid and components connected between the grid nodes, with the values or labels annotated alongside the component symbols. The grid structures of synthetic data are randomly sampled from a predefined hierarchical distribution, ensuring the diversity of shapes, components, and annotations in the generated circuits. We use CircuitTikz as our plotting language to draw the circuit diagram using a LaTeX compiler. Since the component annotations in the real-world diagrams can be in a numerical format (e.g. $10\Omega$) or a label format (e.g. $R_1$), we generate two types of circuit diagrams to cope with this variation accordingly: (1) Numerical-type circuit, where the value is annotated on the diagram. The PPM is required to infer both the type and value of the components; (2) Label-type circuit, where only the labels of the components are provided in the diagram. The PPM predicts the type and label of the components, with an `<Empty>` token in the value position.

The physical structure of a circuit diagram can be represented using by a netlist model (Nagel, 1975; Tao et al., 2024), which is a directed graph where each node represents an equipotential point, and each edge represents a circuit component. The SL synthesis step involves writing rules to automatically identify equivalent circuit nodes using basic physical properties and converting grid information into a netlist. We utilize SPICE (Nagel, 1975) as our simulation language for circuit analysis problems. The syntax of SPICE is based on a netlist model, allowing us directly translated the netlist model into SPICE (Nagel, 1975) program at the end of each generation process. Please refer to Appendix C.1 for our design of the hierarchical distribution and a detailed illustration of the synthesis process.

We name our synthetic data `ppm-syn-lprc`, as our current data synthesis process only supports the generation of Linear Pure Resistive Circuits (LPRC) (Svoboda & Dorf, 2013). `ppm-syn-lprc` contains 20k pairs of synthetic circuit diagrams and their simulation descriptions, divided into training, validation, and test sets in a ratio of 8:1:1.

**PPM Training**. For the training of PPM, we adopt CogVLM-17B (Wang et al., 2023a) as the base model of PPM. The PPM is fine-tuned to generate the SPICE description for given the circuit diagram. For our detailed settings, please refer to Appendix C. Based on our preliminary experiments, the base model is largely unable to accurately perform the conversion task for most circuit diagrams when using prompting methods. Therefore, the training stage is essential for the development of the MAPS pipeline.

**Inference**. In our main experiments, we use GPT-4V as our MLLM and NgSPICE (Nenzi & Vogt, 2011) as our physical simulator to execute circuit simulation. Given a circuit analysis problem with a diagram and textual description, our framework infers the answer to the problem following the process described in Algorithm 1. For more implementation details, please refer to Appendix B.

**Evaluation Dataset**. To evaluate the entire MAPS framework on real-world physical problems, we collected 79 high-quality circuit analysis problems from related textbooks and name it `SimpleCircuitEval`. `SimpleCircuitEval` is constrcuted based on exercise problems primarily collected Chinese circuit analysis text books, but since current MLLMs are primarily multi-lingual and the linguistic type is not an influencing factor in our framework, this should not affect the evaluation of different MLLMs on this dataset. As each question in `SimpleCircuitEval` has an exact golden answer, we can directly compare the answer produced by the candidate model with the golden answer to compute the accuracy. For a fair evaluation of our proposed solution framework, we only retained questions that involve LPRC type questions, which are covered in the first four chapters of the textbook.

## 4.2 EVALUATION OF PPM

We first assess the quality of PPM in translating circuit diagram into SPICE language. We adopt 3 metrics to measure its quality:

**Component Quantity Accuracy ($ACC_{CQ}$).** This metric measures the accuracy of PMM's prediction in terms of the number of circuit components. The prediction is marked as correct only when the number of different types of components are all correct. This measures the object recognition quality of PPM and is a necessary condition for correct conversion from a circuit diagram to its simulation language description.

**Component Value Accuracy ($ACC_{CV}$).** Based on $ACC_{CQ}$, $ACC_{CV}$ requires the model to predict the correct value of each component. This is also a necessary condition and is only applicable for Numerical-Type Circuits. $ACC_{CV}$ reflects both the object recognition quality for circuit components and the PPM's ability to recognize numerical values in the diagram.

**Simulation Accuracy ($ACC_{sim}$).** This metric measures correctness of PPM's conversion results by comparing the consistency of simulation results between the generated SPICE code and the label code. Although $ACC_{CQ}$ is a necessary condition for PPM to be useful in MAPS, in practice, achieving the same simulation results indicates the same physical circuit with high probability. For the specific examples of these metrics, please refer to Appendix C.2.

We first evaluate PPM on the test split of `ppm-syn-lprc`. Then, we integrate PPM into the inference framework for further evaluation on real-world diagrams. The evaluation result of PPM is shown on Table 1. Through training, our PPM can successfully convert most of the synthetic diagrams. For the conversion of real-world schematics, our PPM only has around 50% simulation accuracy, which leaves a big room for further improvement. We provide more in-depth discussions about PPM in Section 5.2.

Table 1: Conversion efficacy of PPM on synthetic dataset `ppm-syn-lprc-test` and 20 diagrams on real-world dataset `SimpleCircuitEval`. "Num." indicates Numerical-Type diagram while "Lab." indicates Label-Type.

| Metrics | ppm-syn-lprc-test | | SimpleCircuitEval | |
|---|---|---|---|---|
| | Num. | Lab. | Num. | Lab. |
| $ACC_{CQ}\uparrow$ | 99.2 | 98.5 | 87.0 | 80.0 |
| $ACC_{CV}\uparrow$ | 95.5 | - | 87.0 | - |
| $ACC_{sim}\uparrow$ | 85.4 | - | 53.3 | - |

## 4.3 EVALUATION OF MAPS FRAMEWORK

To verify the effectiveness of the MAPS framework, we implemented it using existing advanced MLLMs, including GPT-4V (Achiam et al., 2023), Claude-3.5 (Anthropic, 2024) and GLM-4V (GLM et al., 2024). We compared our method with directly prompting these MLLMs to generate the results. Additionally, we implemented the Multimodal-CoT (Zhang et al., 2023b), which prompts the model to generate detailed descriptions and analyses of the given circuit diagram and then infer the result based on the generated multi-modal thought for comparison.

Our main results are reported in Table 2, which demonstrates that MAPS significantly improves the MLLM's multi-modal reasoning capability on circuit-analysis problems and help it outperform existing models and methods. For example, the state-of-the-art GPT-4V only achieved less than 7.6% accuracy on the real-world circuit analysis problems, while our solution raised bar more than 3 times to 32.9%. Through our case studies, we found MAPS effectively alleviates the issues on physical diagrams understanding and complex mathematical reasoning of current MLLM mentioned in Section 2.

We found that our framework and baseline methods all fail at solving problems collected from the Chapter 2 of the textbook, which mainly focuses on the Equivalent Transformation of Resistance and mostly cover the circuits that could not be directly executed in a simulator. When the problem is not simulatable, the MLLM can leverage the additional information from simulation language to reason the final answer. Please refer to our Appendix D.2 for more specific case studies. However, how to improve MLLM's general scientific reasoning ability through interaction with a physical simulator is still a challenging problem and remains for our future work.

Table 2: Evaluation results of MAPS and other baselines on `SimpleCircuitEval`. MAPS significantly surpass existing models and method on complex circuit analysis problems.

| Chapter | ↑ Acc.(%) | | | | |
|---|---|---|---|---|---|
| | Chap1 | Chap2 | Chap3 | Chap4 | All |
| #Problem | 25 | 24 | 19 | 11 | 79 |
| GPT-4V | 16.0 | 4.17 | 5.26 | 9.09 | 7.6 |
| GPT-4V + MMCoT | 8.0 | 4.17 | 5.26 | 0.0 | 5.1 |
| GPT-4V + MAPS (**Ours**) | 52.0 | 7.14 | 36.8 | 45.5 | 32.9(×4.3) |
| Claude-3.5 | 16.0 | 4.17 | 0.0 | 0.0 | 6.33 |
| Claude-3.5 + MAPS (**Ours**) | 44.0 | 12.5 | 21.1 | 36.4 | 25.3(×4.0) |
| GLM-4V | 8.0 | 4.17 | 5.26 | 0.0 | 6.33 |
| GLM-4V + MAPS (**Ours**) | 32.0 | 0.0 | 15.8 | 36.4 | 19.0(×3.0) |
| Gemini-1.5 | 8.0 | 12.5 | 0.0 | 0.0 | 6.33 |
| GPT-4o | 20.0 | 4.17 | 5.26 | 9.09 | 10.1 |

## 5 ANALYSIS

### 5.1 ANALYSIS ON INFERENCE PHASE DESIGN OF MAPS

We perform in-depth analysis of our framework and investigate the contribution of its different components. Our ablation study was performed using a sample of 20 randomly selected problems from `SimpleCircuitEval`. We analyze our MAPS framework by answering a series of questions.

***Q: Can we directly prompt MLLM to generate simulation language descriptions of given circuit diagrams, instead of training the expert model PPM?***

***A***: Despite being pre-trained on large-scale corpora, we found that even the most advanced MLLMs, such as GPT-4V, often struggle to generate accurate simulation language descriptions for relatively complex circuit diagrams. Specifically, GPT-4V refused to generate SPICE code in 8 out of 10 instances in our evaluation set.

***Q: Is the simulator necessary for MAPS framework?***

***A***: We use ablation analysis to answer this question and report the results in Table 3. We found that MAPS does not work without the assistance of the simulator when we remove the simulation results from the final query (i.e., **MAPS w.o. Simulator**). We also verified the necessity of simulator by prompting MLLM to write Python programs to infer the answer (Chen et al., 2022) given the simulation language and problem description (i.e.,

Table 3: Ablation study of MAPS framework on Problems sampled from `SimpleCircuitEval`. The results show a high reliance of the MAPS framework on the physical simulator.

| Method | ↑ Acc.(%) |
|---|---|
| MAPS (**Ours**) | **55.0** |
| MAPS w.o. SL | 45.0 |
| MAPS w.o. Simulator | 15.0 |
| MAPS w.o. Simulator + PoT | 15.0 |

**MAPS w.o. Simulator + PoT**). Notably, **MAPS w.o. Simulator** and **MAPS w.o. Simulator + PoT** both achieved only 15% accuracy on the evaluation set. This underscores the importance of incorporating a professional simulator when addressing problems with complex physical backgrounds.

***Q: Is the simulation language description helpful for the final reasoning?***

***A***: We found that when the problem is not simulatable, the simulation language can still be helpful to the final reasoning of framework. The structural information provided by the simulation language significant reduces hallucination of MLLM when understanding the diagram, akin to the role of scene graph in general multi-modal reasoning (Mitra et al., 2024). Appendix D.2 presents a detailed example showing how simulation description in MAPS alleviate the MLLM's hallucination problem when the physical scene is not simulatable.

We also investigate whether the simulation language description is necessary to simulation-aided reasoning step when simulation results are given, denoted as **MAPS w.o. SL** in Table 3. The result shows that the SL plays a vital role in final reasoning even when the simulation results are given, bridging the gap between the diagram information and numerical simulation results.

## 5.2 ANALYSIS OF PPM CONSTRUCTION

**Philosophy of PPM Construction**. Although we focus solely on circuit disciplines in our evaluation, the philosophy of constructing a PPM is universal across all physical disciplines. The target of PPM is to convert a physical diagram to its **formal** and **simulatable** language description, which requires paired training data in the form of physical diagrams and corresponding simulation language descriptions.

Since there is no available open-source data in such a format and human annotations on a large corpus is quite costly, we devised an automated data synthesis solution to enhance the VLM's perception ability on real-world diagrams. The assumption behind our data synthesis pipeline is that *the potential space of physical diagrams can be effectively covered by a human-designed distribution*. Physical diagrams are often composed of dots, lines, and symbols with specific physical meanings, and are primarily designed to abstract real-world scenarios. By distilling the core patterns of these diagrams, we can establish a distribution to generate representative training data for PPM. For example, in circuit diagrams, we have observed that most inputs are formed with planar grids, with components placed at the edges of these grids. In mechanical diagrams, the pattern could be composition and positional relationship of mechanical objects (pole, ball, box etc.). Since our work is exploratory, designing a universal generator for physical diagrams or obtaining a comprehensive physical perception model remains an open problem.

**Using VLM to implement PPM**. Converting a physical diagram into its simulation language description can be viewed as a comprehensive vision task which involves the recognition of physical objects and the OCR of its detached labels, along with the complex topology about the components' connection. In terms of circuit schematics, some previous works (Bayer et al., 2023; Bailey et al., 1995; Tao et al., 2024) investigate multi-step process to convert a pixel-level circuit into digital structure, but their methods are expensive to implement and not scalable to diagrams of new styles.

By using VLM as our perception model, we obtain an end-to-end physical diagram recognition solution whose capability can be extended through expanding the data distribution during training. Besides, we also observe that pre-trained VLMs exhibit promising generalization ability after training on our synthetic data, e.g., its OCR ability on float number although our synthetic data only contains integer values.

**Scaling the conversion task**. Through a development set based on our synthetic Numerical-Type Circuits, we also found that the conversion accuracy ($ACC_{sim}$) decreases as circuit's complexity increases. Figure 3 illustrates that as the number of nodes and components increases in our synthetic data, the simulation accuracy of PPM's predictions shows a downward trend. This result is intuitive since the smaller circuits with simpler physical structures show higher accuracy during test.

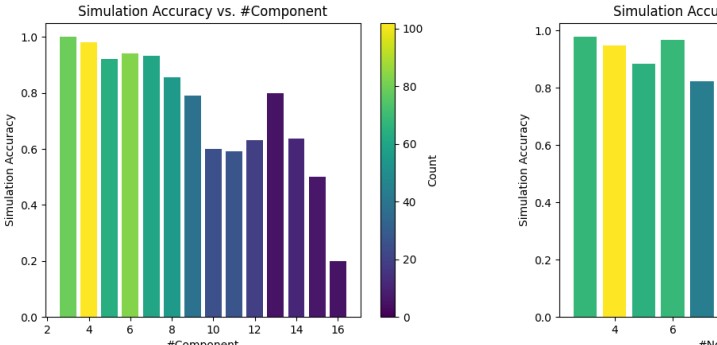

Figure 3: With the increase in circuit scale—specifically the number of components and electrical nodes—the accuracy of PPM decreases. The colorbar display the sample amount in each scale.

## 6 DISCUSSION & CONCLUSION

In this work, we introduce the MAPS framework to address the inability of existing MLLMs in understanding complex physical diagrams and to solve such problems analytically. Our framework, which trains a Physics Perception Model (PPM) to interpret physical diagrams and applies Chain-

of-Simulation and Simulation-Aided Reasoning during inference, successfully solves the circuits analysis problem, a typical and important type of real-world physical problems.

MAPS excels in deriving final answers when the physical scenario is directly simulatable. However, a key limitation is its static workflow, which lacks feedback interaction with the physical simulator. To address this, a dynamic workflow where the simulator acts as an external environment for feedback is necessary. In this setting, PPM still serves an important role in connecting multi-modal information with the physical simulator. This improvement would significantly enhance the versatility of our physical agent and is an important focus for future work.

As the first attempt of this kind, this work only tested MAPS on LPRC circuit analysis problems. Extending MAPS to other scientific disciplines with complex illustrative schematics is an important next step. It requires developing a universal and accurate PPM for the Chain-of-Simulation. This is a challenging task in computer vision that remains for future work. Additionally, simulators are currently domain-specific, making effective organization across simulators of different domains or the development of a universal simulator crucial for MAPS's broader application.

As shown in our experiment results, our work presents a solid path towards building multi-modal agents capable of solving expert-level scientific problems, contributing to the progress towards achieving AGI in scientific domains.

## 7 ACKNOWLEDGEMENT

This work was supported by the National Science Foundation for Distinguished Young Scholars (with No. 62125604) and Tsinghua University Initiative Scientific Research Program.

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

# A    RELATED WORK

**Improving Multi-modal Reasoning Ability of MLLM.** Reasoning ability is foundational for building agent that assist human to solve complex real-world tasks. There are many studies focusing on improving the reasoning ablility of MLLMs. There have been three main directions to boost the reasoning capability of MLLMs, including instruction-tuning, prompt engineering and tool use (Wang et al., 2024c). As instruction tuning requires high-quality multi-modal training corpus which is scare in scientific domains, most studies focus on how to improve the scientific reasoning ability of MLLMs via prompting methods and tool utilization.

The multi-modal prompting methods involve designing effective prompts to fully activate the model's image understanding and language reasoning abilities based on the carefully crafted text instructions. Specifically, existing methods (Zhang et al., 2023b; Mitra et al., 2024; Zheng et al., 2023; Zhou et al., 2024; Zhong et al., 2024; Yang et al., 2023) focus on how to enable the model to generate intermediate rationales, or chain-of-thought (CoT) (Wei et al., 2022), for parsing image information and then further reason based on these intermediate results (Zhang et al., 2023b). Some variants of Multimodal-CoT focus on the form of CoT, for example, Zheng et al. (2023) format the CoT as a decomposition of original problems and use the answers of sub-problems to generate the result, while Mitra et al. (2024) adopt Scene Graph (SG) as the rationale to assist the inference of final answer.

On the other hand, integrating LLMs with external tools exhibits significant improvement in scientific reasoning (Chen et al., 2022; Gou et al., 2023). For the reasoning problems containing data from different modalities, the category of available tools extends beyond traditional external software (e.g., calculators, calendars, search engines, code execution environments) to specialized vision models (e.g., object detection models, OCR models, image semantic segmentation models, etc.). Research in this direction seeks to build useful tool sets and design mechanisms for MLLMs to interact with these tools, thereby completing designated reasoning tasks more effectively (Liu et al., 2023; Wang et al., 2024a; Gao et al., 2023).

**Multi-Modal Agent in Scientific Scenarios.** A multi-modal AI agent is a system designed to realize users' general-purpose requirements by perceiving its environment with multi-modal information and making decisions based on its observations (Xie et al., 2024). Multi-modal agents have been developed for many important domains, including GUI automation (Gur et al., 2024; Wen et al., 2024; Wang et al., 2024b), embodied AI (Qin et al., 2024; Wang et al., 2023b) and general understanding, generation and editing of images, videos and audios (Wu et al., 2023; Gao et al., 2023; Liu et al., 2023; Yang et al., 2023). For example, Gur et al. (2024) devise an LLM-based agent that learns from interactive experiences to follow human instructions and complete tasks on real websites, such as click, type or making selection.

To construct multi-modal agents in scientific domains, an important research direction involves how to perceive multi-modal information encompassing diverse scientific concepts. With the development of LLMs, a lot of efforts have been made to build multi-modal foundation models tailored for specific scientific scenarios. These models are capable of perceiving or generating chemical formulas, protein sequences, geographical information, graphs, and more (Luo et al., 2022; Li et al., 2023; Frey et al., 2023; Jiang et al., 2023; Zhang et al., 2023a). However, most previous works focus solely on multi-modal information in text format, neglecting the pixel-format information of physical diagrams that are prevalent in human knowledge bases.

# B    ADDITIONAL IMPLEMENTATION DETAILS OF MAPS INFERENCE PHASE

## B.1    SIMULATION

For the simulation of circuit problems, we use NgSPICE (Nenzi & Vogt, 2011) developed by the UC Berkeley CAD Group as our simulator. The core arguments we set for the simulation are listing on Table 4.

We store the simulation results in `dictionary` format, which is a commonly used data structure in programming as well as the conversation with MLLM (Mitra et al., 2024; Weber, 2024). Figure 4 shows an example of the post-processing of simulation result.

Table 4: Parameters of NgSPICE simulator

| Param. | Setting |
| --- | --- |
| Temperature | 27° |
| Nominal Temperature | 27° |

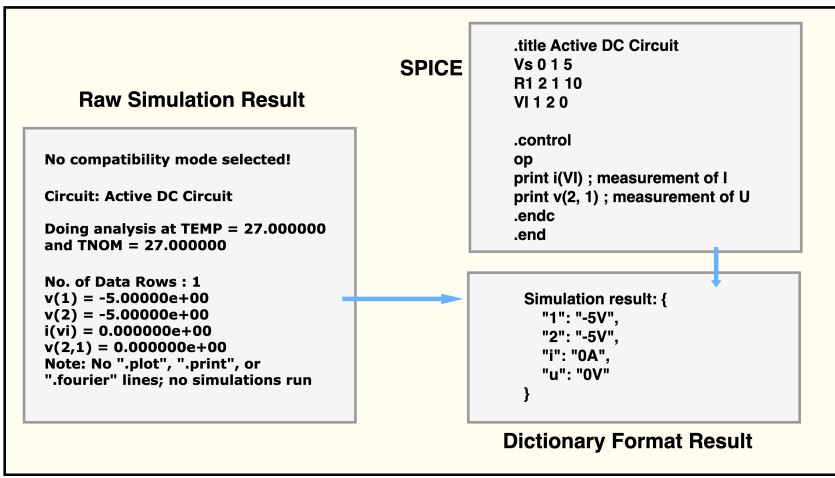

Figure 4: Post-processing of simulation results.

### B.2 PROMPT TEMPLATES

In this section, we will showcase the prompt templates that we used at Inference Stage.

At the Chain-of-Simulation step of MAPS inference, since our training PPM is merely an image-to-text model, the component values of circuit in textual description is merged to the simulation language by MLLM in Refine process.

The prompt we used for this process is shown at Figure 5 . This prompt will only be applied when we detect `<Empty>` token in generated simulation language of PPM, which is a special token design for the component with missing value in the diagram.

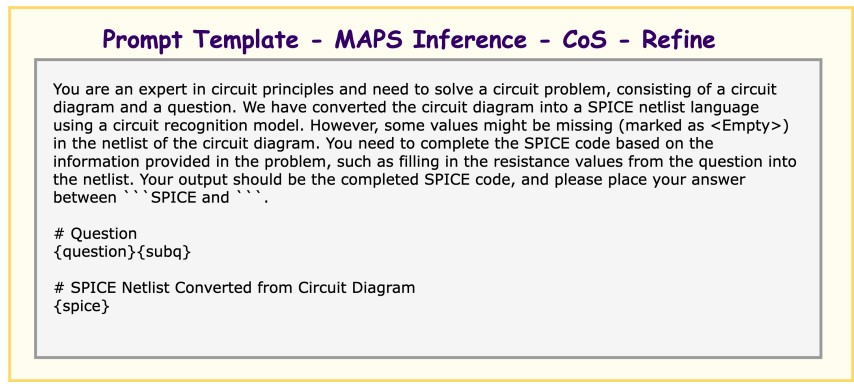

Figure 5: Prompt template of refine process at Chain-of-Simulation step

In the Simulation-Aided Reasoning(SAR) step, the MLLM infers the answer based on the information provided by Chain-of-Simulation. Figure 6 shows the prompt template used for SAR in circuit disciplines.

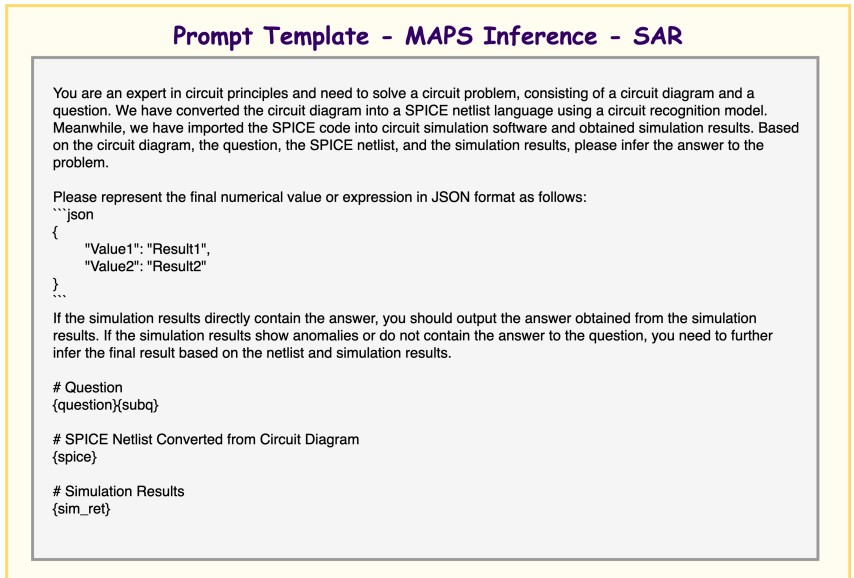

Figure 6: Prompt template of Simulation-Aided Reasoning step

If the simulation results are not obtained (due to incorrect simulation language) in SAR step, we use a special prompt that allows the MLLM to infer the final result based on the information provided in the problem and the simulation language. Figure 7 shows the special prompt.

**Prompt Template - MAPS Inference - SAR w.o. simulation result**

You are an expert in circuit principles and need to solve a circuit problem, consisting of a circuit diagram and a question. We have converted the circuit diagram into SPICE netlist language using a circuit recognition model. Based on the question, the circuit diagram, and the SPICE netlist, please infer the answer to the problem.

Please represent the final numerical value or expression in JSON format as follows:
```json
{
    "Value1": "Result1",
    "Value2": "Result2"
}
```

# Question
{question}{subq}

# SPICE Netlist Converted from Circuit Diagram
{spice}

Figure 7: Prompt template of Simulation-Aided Reasoning step (No Simulation Result)

Figure 8 shows the prompt template that we prompt MLLM to directly infer the answer. Since current MLLMs have been trained on CoT data, they will apply an automated CoT to infer the answer.

MMCoT (Zhang et al., 2023b) decomposes the multi-modal reasoning process into a two step paradigm: Rationale Genetraion and Answer Inference. Following the similar idea, we define the rationale in our setting as the language description of the physical diagrams. Figure 9 and 10 display our prompt templates for the two step generation.

**Prompt Template - MLLM Direct**

```
Please solve the following circuit problem, and represent the final numerical values or expressions
in JSON format, formatted as:
```json
{
     "Value1": "Result1",
     "Value2": "Result2"
}``

# Question
{question}{subq}

# Accompanying Diagram
{diagram}
```

Figure 8: Prompt template of MLLM directly inference

**Prompt Template - MLLM + MMCoT Step 1**

```
You will receive a circuit problem with a circuit diagram. You need to generate a descriptive language
for the circuit diagram to facilitate further inference. {extra_prompt}

# Question
{question}{subq}

# Accompanying Diagram
{diagram}
```

Figure 9: Prompt Template for Step 1 of MMCoT

**Prompt Template - MLLM + MMCoT Step 2**

```
You need to solve a circuit problem with a circuit diagram, and you can refer to the descriptive
language of the circuit diagram. Please represent the final numerical values or expressions in JSON
format, formatted as:
```json
{
     "Value1": "Result1",
     "Value2": "Result2"
}``

# Question
{question}{subq}

# Circuit Diagram Descriptive Language
{desc}

# Accompanying Diagram
{diagram}
```

Figure 10: Prompt Template for Step 2 of MMCoT

# C ADDITIONAL IMPLEMENTATION DETAILS OF PPM CONSTRUCTION PHASE

In this section, we will delve into the implementation details involved in the construction phase of the Physics Perception Model (PPM).

## C.1 DATA SYNTHESIS

The data synthetic pipeline has been shown in the left side of Figure 2(a). We have introduced the general process of data generation in Section 3.2.1. In this section, we will introduce our synthesis pipeline in circuit discipline with a specific example.

In the first step, we sample an diagram layout from the manual distribution. As discussed in Section 5.2, the key property of our designed generation distribution is to cover the distribution of real-world diagrams as comprehensively as possible.

Our implementation of diagram layout sampling in synthesizing PPM's training data for Linear Pure Resistive Circuit (LPRC) (Svoboda & Dorf, 2013) diagrams is shown in Algorithm 2, where $\mathcal{D}, \mathcal{U}$ in the pseudo code represent Discrete Probability Distribution and Uniform Distribution respectively. We only show our main idea in the pseudo code due to its tedium. Since we use a hierarchical sampling process, we can sample diverse circuits with different shapes, components and annotations. The hyperparameters of the sampling process are set by human experiences.

---

**Algorithm 2** Diagram Layout Sampling for LPRC

1: **Input:** $d_{\max}, d_{\min}, \vec{n}, \vec{p_n}, \vec{t}, \vec{p_t}, \cdots$
2: **Output:** Diagram Layout $\mathcal{I}$
    % Determine the scale of the grid
3: number of grid: $(m \times n)$: $m, n \sim \mathcal{D}(\vec{n}, \vec{p_n})$
4: horizontal scale: $\vec{d^h} = (d_1^h, \cdots, d_n^h)$ where $d_i^h \sim \mathcal{U}(d_{\min}, d_{\max})$
5: vertical scale: $\vec{d^v} = (d_1^v, \cdots, d_m^v)$ where $d_i^v \sim \mathcal{U}(d_{\min}, d_{\max})$

    % Determine the component's type & direction in each edge
6: horizontal component: $T^h = [T_{i,j}^h]_{m \times (n-1)}$ where $T_{i,j}^h \sim \mathcal{D}(\vec{t}, \vec{p_t})$
7: vertical component: $T^v = [T_{i,j}^v]_{(m-1) \times (n)}$ where $T_{i,j}^v \sim \mathcal{D}(\vec{t}, \vec{p_t})$
8: direction of horizontal component: $D^h = [D_{i,j}^h]_{m \times (n-1)}$ where $D_{i,j}^h \sim \mathcal{D}((0,1), (0.5, 0.5))$
9: direction of vertical component: $D^v = [D_{i,j}^v]_{(m-1) \times n}$ where $D_{i,j}^v \sim \mathcal{D}((0,1), (0.5, 0.5))$

    % Determine the component's value & unit in each edge
10:    $\cdots$

    % Determine the component's label in each edge
11:    $\cdots$

    % Determine the observation's label & direction in each edge
12:    $\cdots$

    % Assign the observation label to controlled source
13:    $\cdots$
14: $\mathcal{I} = \text{CircuitDiagram}(m, n, \vec{d^h}, \vec{d^v}, ...)$
15: **return** $\mathcal{I}$

---

In our illustrative case, we sampled a $4 \times 4$ grid and assign each edge with specific components, as shown in Figure 11.

After the sampling of diagram layout, there are two synthesis paths that respectively generate pixel-format diagram and simulation language(SL) description.

The diagram synthesis path involves converting the grid into LaTeX language that can describe a circuit diagram. We use the LaTeX package `circuitikz` to plot the circuit. After compiling the LaTeX code, a pixel-level circuit diagram can be generated. Each edge in the grid is converted into a line of LaTeX drawing language. The drawing statement includes the start and end positions of

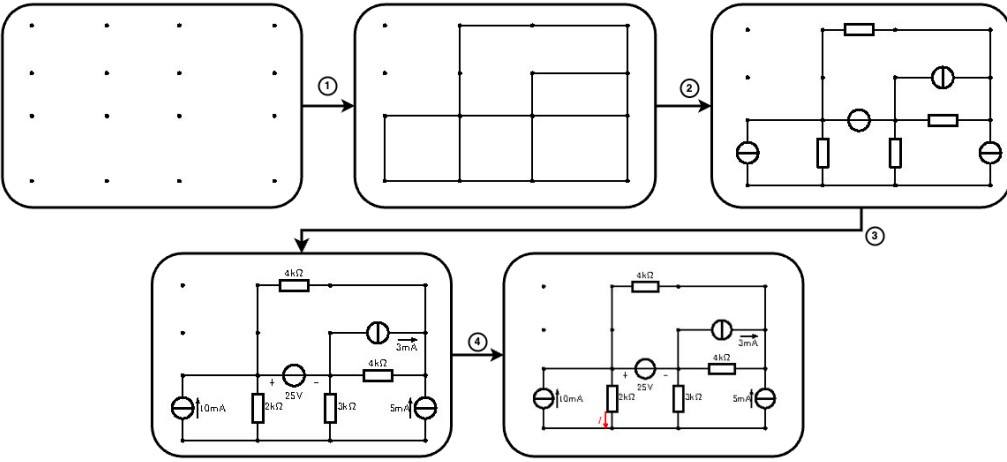

Figure 11: Data Synthesis: Diagram Layout Sampling

the element/wire, the shape of the element, the label of the element (number or string), the type of measurement, and its label.

The SL synthesis path primarily focuses on distilling the physical structure from the diagram layout using human prior knowledge. The physical structure of a circuit can be represented by a netlist (Nagel, 1975; Tao et al., 2024) model, which is a directed graph (West et al., 2001) where each node represents an equipotential point and each edge represents a component. The netlist model includes all components along with their types, parameters, and topological connections of a circuit, while filtering out position and scale noise when plotting the diagram. We write rules to automatically identify equivalent electrical nodes using basic physical properties and convert grid information into a netlist. Figure 12 illustrates the physical structure extraction process of our example case.

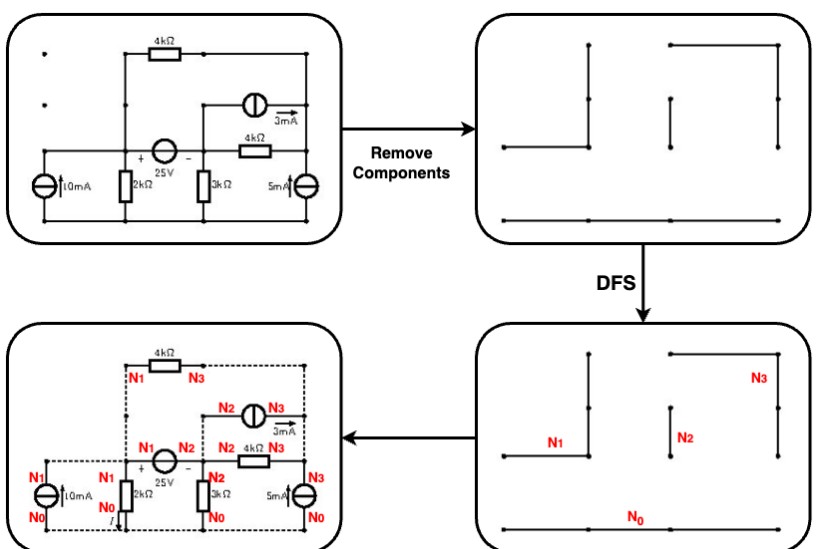

Figure 12: Data Synthesis: Extracting physical structure from diagram layout using physical rules

Finally, the netlist is translated into SPICE code, i.e. our simulation language as the training label of PPM. The SPICE language can be mainly divided into two parts: the first part is the description of circuit elements (Element Card), and the second part consists of control commands that determine the simulation type and output results (Control Card). Since each edge in the netlist corresponds

directly to a circuit element description line in SPICE, the conversion of the first part is merely a formatting process. For the second part, which involves control commands, we set up the simulation for a steady-state analysis of Linear Purely Resistive Circuits(LRPC). This is done by using the .OP (Operating Point) command to define the simulation type as a DC operating point analysis and the .PRINT command to specify the circuit state quantities to be measured.

We also counted the amount of electrical nodes and components in our synthetic dataset `ppm-syn-lprc-test`. The statistic results are shown on Table 5, where "#X" represents the number of objects of type X.

Table 5: Statistics of `ppm-syn-lprc-test`

| Parameter | Mean | Std | Max | Min |
|---|---|---|---|---|
| **#Nodes** | 7.876 | 3.137 | 26.0 | 1.0 |
| **#Branches** | 11.088 | 5.364 | 45.0 | 0.0 |
| **#Resistors** | 6.566 | 3.452 | 25.0 | 0.0 |
| **#Voltage Sources** | 1.340 | 1.268 | 9.0 | 0.0 |
| **#Current Sources** | 1.517 | 1.340 | 12.0 | 0.0 |
| **#Controlled Sources** | 1.411 | 1.457 | 11.0 | 0.0 |
| **#Shorts** | 0.255 | 0.508 | 5.0 | 0.0 |
| **#Voltage Measurements** | 1.192 | 0.834 | 7.0 | 0.0 |
| **#Current Measurements** | 0.503 | 0.713 | 6.0 | 0.0 |

Figure 13 shows some cases of our synthetic circuit diagrams.

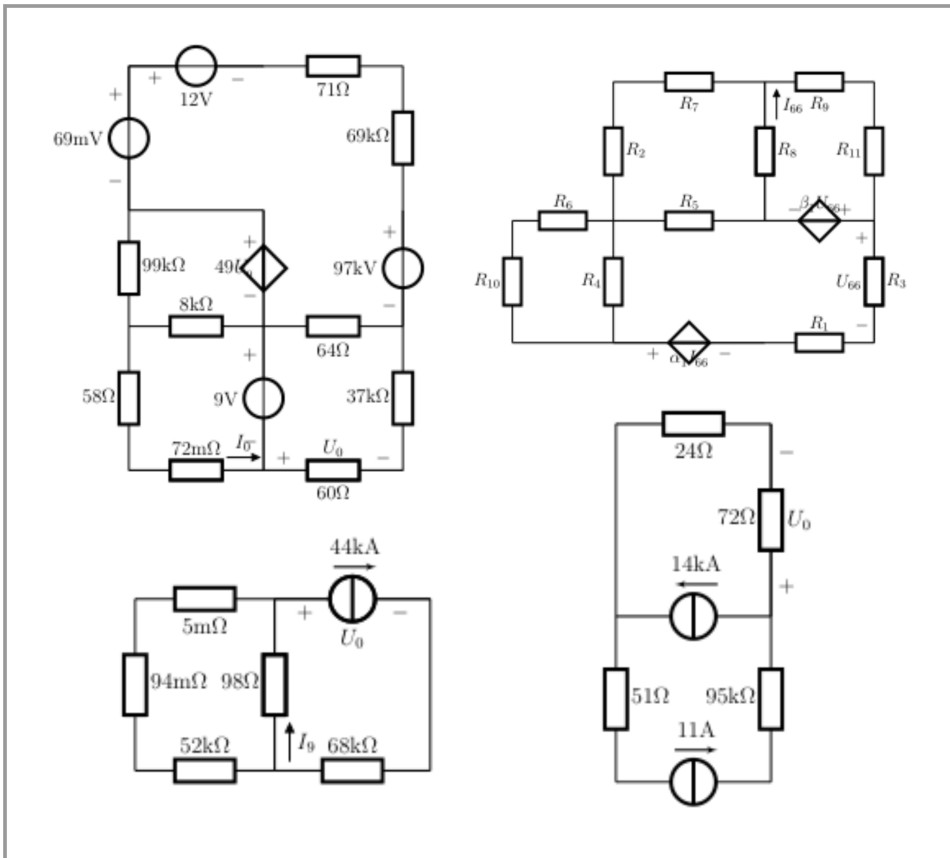

Figure 13: Examples of synthetic diagrams

The synthetic paired data are used for the training of physics perception model.

## C.2 PPM TRAINING

We will introduce our PPM training process for our main experiments in detail in this section.

The training objective of PPM is to predict the simulation language given the visual diagram input. Let the diagram input be $X_V$, and the output text sequence be $Y_L = (y_{L,1}, ..., y_{L,T})^T$, with a length of $T$. The model parameters are denoted as $\theta$. The probability distribution for predicting the next token under the model can be represented by $p_\theta(y|X_V, Y_{L,1:t})$. The MLE fine-tuning loss can therefore be written in the form $\mathcal{L}_\theta^{MLE}(X_V; Y_L) = -\sum_{t=0}^{T-1} \log p_\theta(y = y_{L,t+1}|X_V, Y_{L,1:t})$, where $Y_{L,1:t} = (y_{L,1}, ..., y_{L,t})^T$.

Let the training dataset be $\mathcal{D} = \{X_V^{(i)}; Y_L^{(i)}\}_{i=1:N}$, containing $N$ samples. The training process involves minimizing the negative MLE loss for all training samples:

$$\theta^* = \min_\theta \sum_{X_V^{(i)}, Y_L^{(i)} \sim \mathcal{D}} \mathcal{L}_\theta^{MLE}(X_V^{(i)}, Y_L^{(i)}) \tag{1}$$

In our experiments, we adopt CogVLM-17B as our base model to train the PPM. The model version for main experiment is `cogagent-vqa-17B`.

We primarily train the visual modules and the image-text cross-attention part, while the parameters of the text generation part remain mostly unchanged. This is because the main challenge of this task lies in image understanding, and the text generation aspect has already been adequately learned through pre-training in the language model part of CogVLM. We control the trainable parameters of the model as follows: the visual encoder, the ViT, the visual multi-layer perceptron and rotary encoding module, the BOI token and the EOI token, resulting in a total of 11.6B parameters that need updating. The remaining parameters are kept frozen.

We employ Low-Rank Adaptation (LoRA) (Hu et al., 2021) as the fine-tuning strategy to train the VLM. Using the LoRA algorithm to train the VLM significantly reduces the number of parameters for which gradients need to be computed, thereby greatly decreasing the memory overhead of training.

We list our main hyperparameters used for PPM training at Table 6.

Table 6: Main Hyper-parameters of PPM Training

| Param. | Setting |
|---|---|
| lora-rank | 50 |
| max-length | 2000 |
| batch-size | 32 |
| train-iters | 2000 |
| optimizer | Adam |
| learning-rate | 1e-5 |
| lr-decay-style | cosine |
| warmup | 0.2 |

After the training process, we evaluated the PPM using the metrics introduced in Section 4.2. To illustrate how these metrics work, we present three cases in Figure 14.

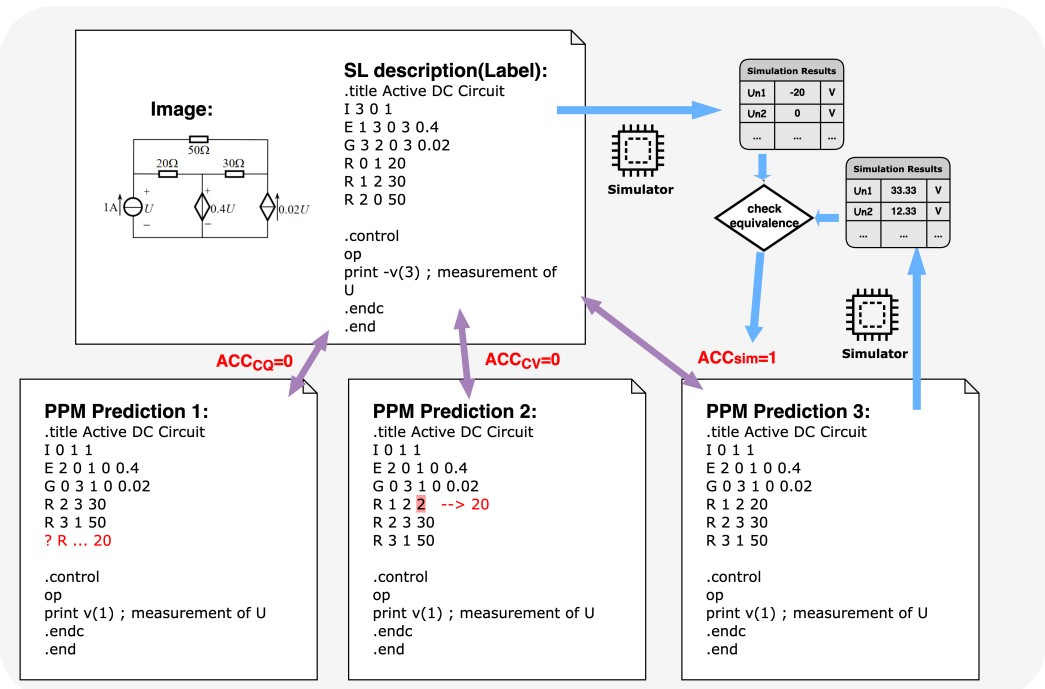

Figure 14: Cases of evaluating PPM.

## D  ADDITIONAL DETAILS OF EVALUATION ON SIMPLECIRCUITEVAL

### D.1  DETAILS OF SIMPLECIRCUITEVAL

To evaluate the performance of MAPS on Linear Pure Resistive Circuits (LPRC), we selected questions involving LPRC from the first four chapters of a circuit course textbook. To ensure question diversity and coverage, we consulted domain experts to remove redundant questions, resulting in a final set of 79 questions. The characteristics of the problems from these four chapters and their details are summarized in the table below:

Table 7: Summarization of SimpleCircuitEval.

| Chapter | Content | #Components(Avg.) | Characteristics |
|---|---|---|---|
| 1 | Circuit elements and circuit laws | 4.7 | Basics circuits. No controlled sources. |
| 2 | Analysis method of simple resistance circuit | 6.8 | Resistance circuits. Not directly simulable, require calculation of equivalent resistance. |
| 3 | General method of analysis of linear resistance circuits | 7.6 | LPRC circuits. Normal topologies. |
| 4 | Some theorems of electric circuits | 6.2 | LPRC circuits. Complex topologies. |

In Figure 15 we offer 4 example question for each chapter in SimpleCircuitEval.

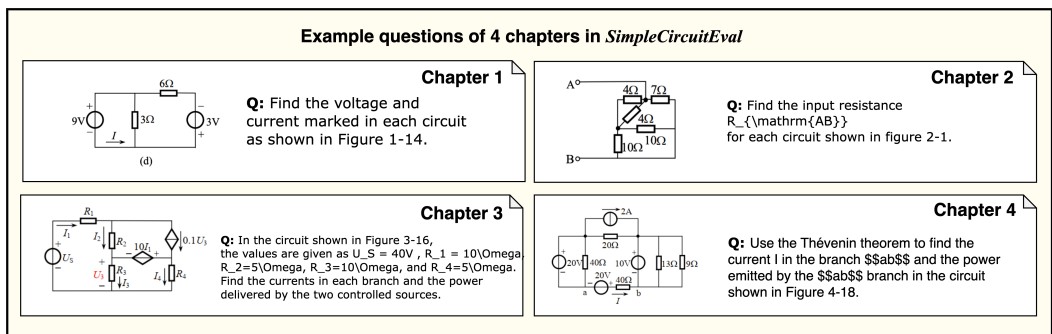

Figure 15: Example questions in `SimpleCircuitEval`.

## D.2 SUCCESSFUL EXAMPLES

We found simulation language can reduce the hallucination of MLLM on understanding the physical diagram, as shown on Figure 16. These results are consistent with those in a concurrent work (Tao et al., 2024), which uses SPICE descriptions as auxiliary information to guide MLLM's decisions in the context of IC design.

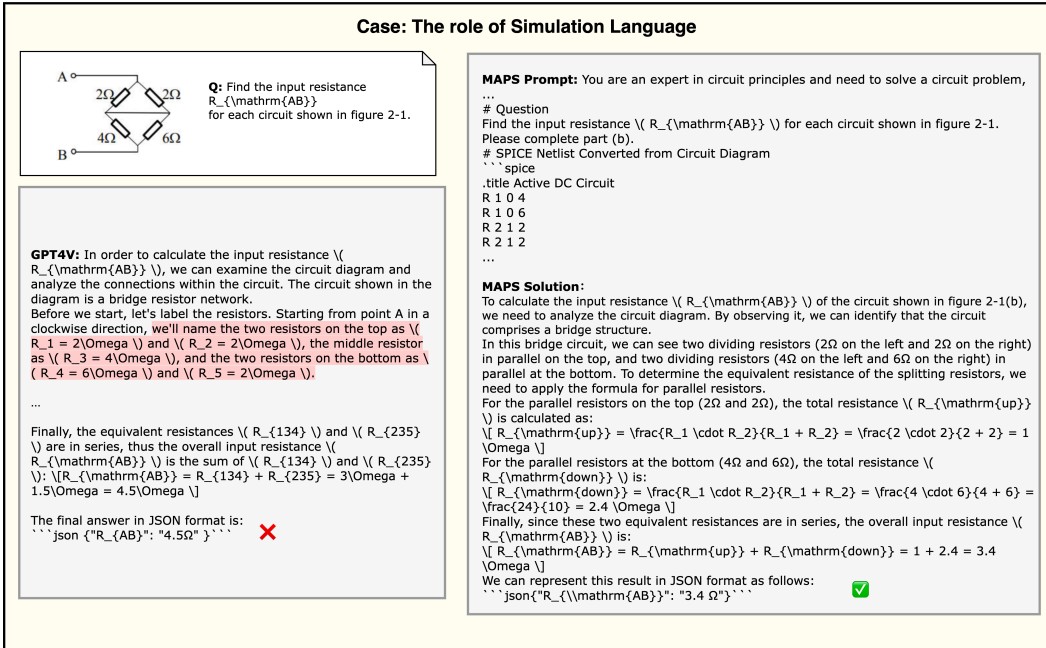

Figure 16: Case Study: The vital role of simulation The positions marked in red are the hallucination positions of the MLLM.

Figure 17 illustrates how MAPS effectively overcomes the challenge of MLLM's inability to comprehend complex physical diagrams by employing formal simulation descriptions and executing precise simulations. In another case, as shown in Figure 18, we found that MAPS successfully addresses the issue of MLLM's inability to perform derivations of complex equations.

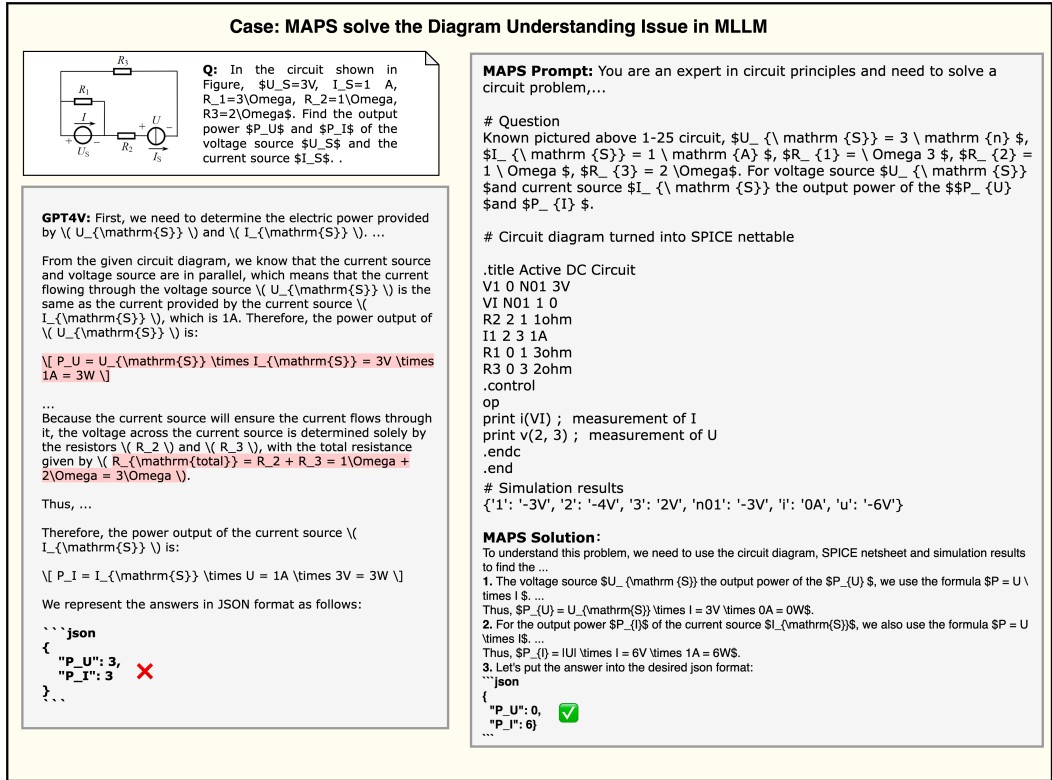

Figure 17: Case Study: MAPS addresses the diagram understanding issue of MLLM.

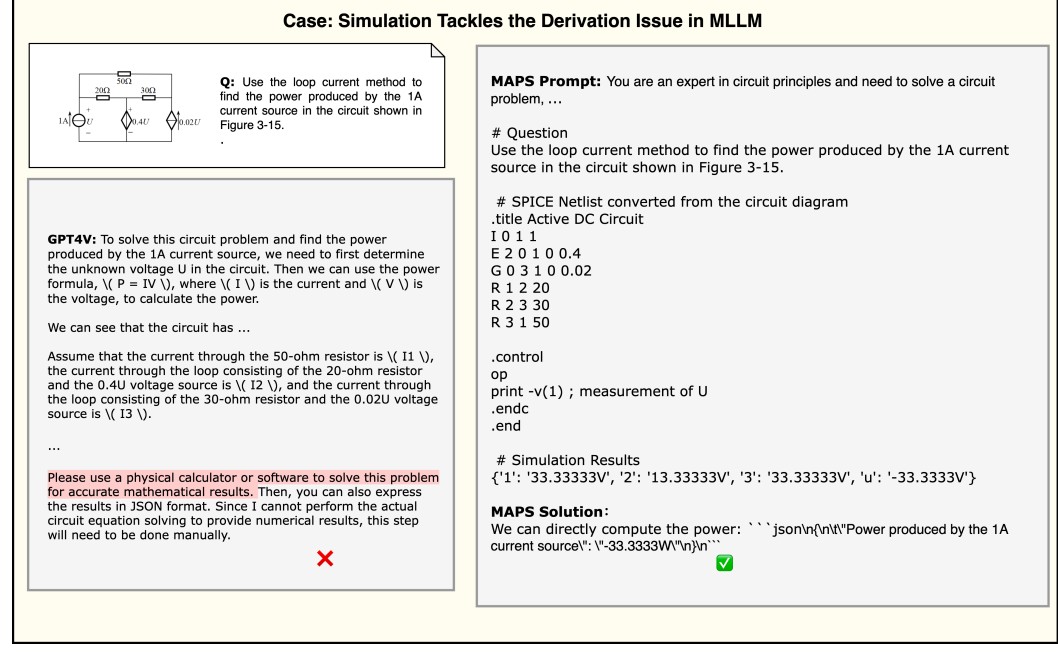

Figure 18: Case Study: MAPS solves the math derivation issue of MLLM.

## D.3 ERROR ANALYSIS

Through the analysis of erroneous samples, we identified two primary causes of errors in MAPS:

1. Incorrect simulation description conversion during the Chain-of-Simulation step. Due to the relatively precise solutions produced by the physical simulator, the errors in Chain-of-Simulation (CoS) process can only occur during the translation step of the simulation language description (SLD). These errors specifically manifest in the incorrect recognition of components by PPM, errors in the identification of circuit topology by PPM, and mistakes in the MLLM refinement process of the SLD. Upon our observation, these types of errors constitute the majority, accounting for 18 out of 20 errors.

2. Hallucination during the Simulation-Aided Reasoning step. We also found that even when the PPM generated a correct simulation language description, the final inference result was still incorrect. This is primarily due to the limited mathematical reasoning capability of the MLLM.

We present two typical cases, shown in Figure 19 (Error in CoS) and Figure 20 (Hallucination in SAR step), where our MAPS framework fails to solve the problem.

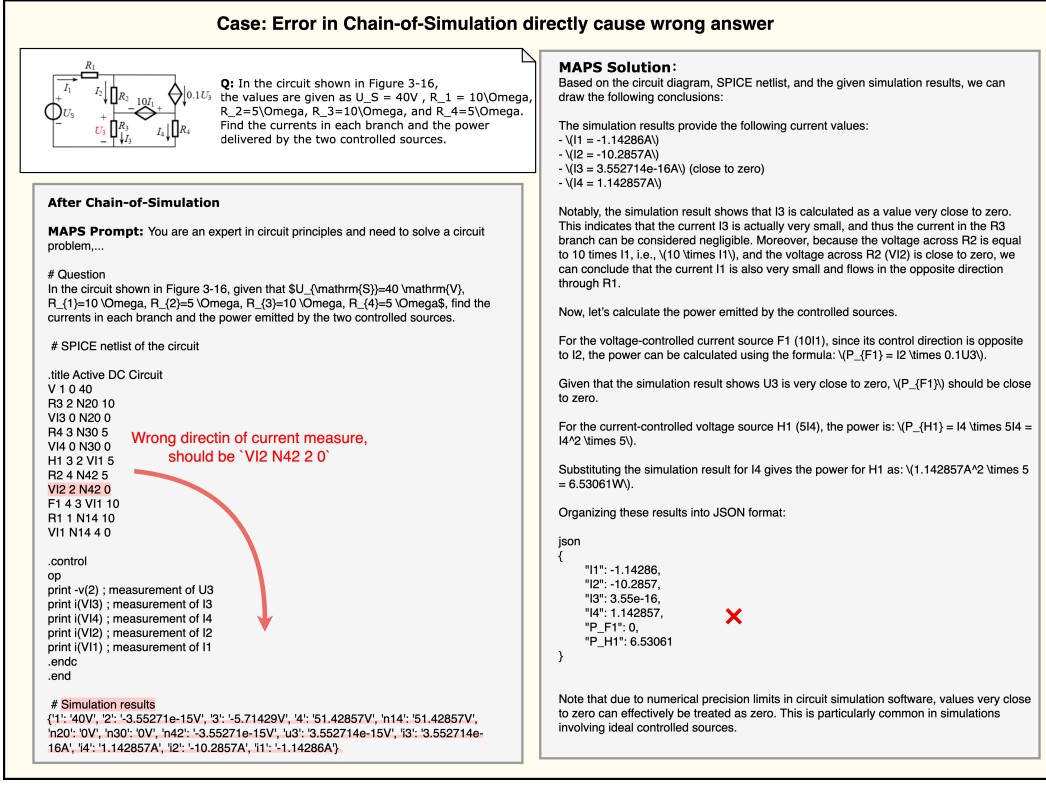

Figure 19: Case Study: Error in Chain-of-Simulation.

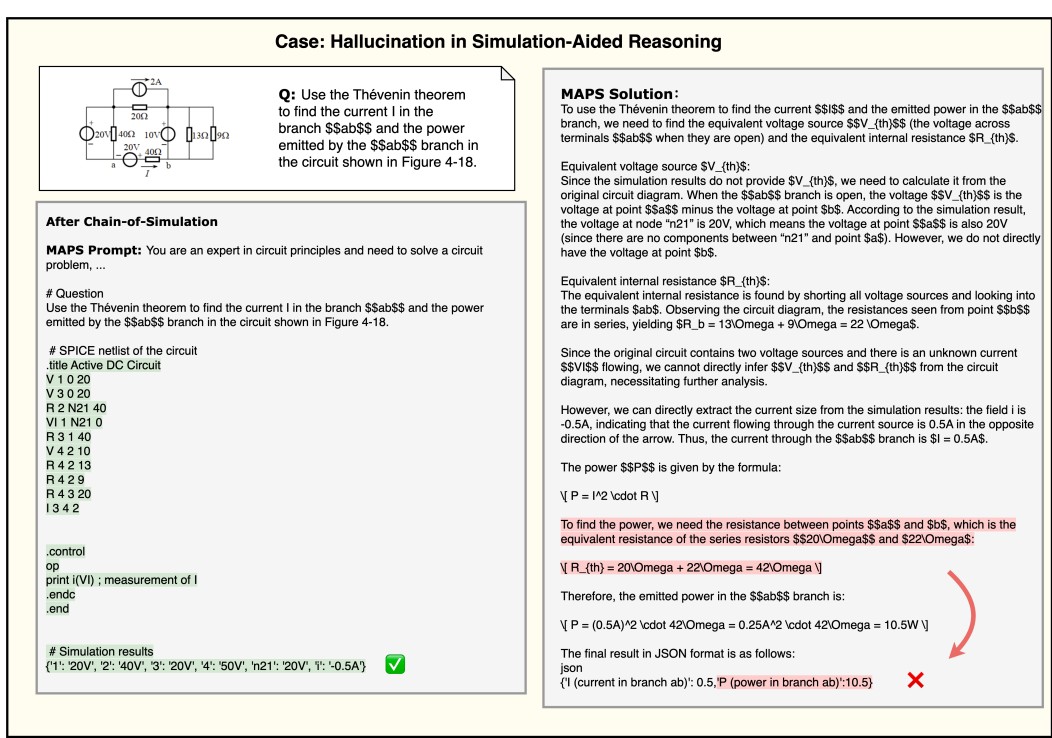

Figure 20: Case Study: Hallucination in Simulation-Aided Reasoning step.

