# OpenReview forum: "MAPS: Advancing Multi-Modal Reasoning in Expert-Level Physical Science"
_ICLR.cc/2025/Conference — ICLR 2025 Poster_

### Official Review · Reviewer_b9qd · 2024-10-26

**Soundness:** 3
**Presentation:** 3
**Contribution:** 2
**Rating:** 6
**Confidence:** 4

**Summary:**

The paper proposes to use improve Multi-Modal LLM's reasoning capability by fine-tuning with domain-specific formal languages and using external physics simulator.

**Strengths:**

1. There is some novelty in augmenting MLLM with the formal languages describing the circuits and using an external physics simulator.
2. The proposed method works well in improving the performance.
3. The paper is written clearly and easy to follow

**Weaknesses:**

1. The paper only proposed a solution for a very specific domain of physical reasoning, namely circuit analysis. This implies that for every other sub-domain, we need to use another set of formal language and physics simulator, which is not scalable. This also implies that this philosophy cannot be extended to physical reasoning problems that don't have any formal languages describing it, which reduces its potential in the frontiers of science research.

**Questions:**

None.

---

> ### Author Response · Authors · 2024-11-21
> **Response to Reviewer b9qd**
>
> We sincerely appreciate the reviewer's insightful feedback. The adaptability of MAPS across diverse scenarios is indeed a crucial consideration.
>
> **[Q1]** Adaptability across different physical sciences, particularly those lacking formal language descriptions.
>
> **[A1]** We address the issue in salability of MAPS comprehensively in **[CA2]** of the Common Response to All Reviewers. Currently, MAPS is mainly designed for well-established disciplines with standardized rules and formal languages. Considering the continuous development of science, there may be emerging disciplines that lack a universal formal language to describe their physical scenarios. For such disciplines, the MAPS framework cannot be quickly adapted at this time. However, as most scientific disciplines evolve, formal language descriptions tend to become established and standardized over time, at which point MAPS can be effectively adapted to these new disciplines.
>
> We welcome any additional questions or suggestions the reviewer may have. We are eager to further improve our work based on the feedback, which could lead to a more favorable evaluation.

---

> > ### Author Response · Authors · 2024-11-27
> >
> > Dear Reviewer b9qd
> >
> > Thank you very much for your insightful feedback. We are greatly encouraged by your recognition of the innovation and clarity of our work. During the rebuttal phase, we provided a detailed explanation for your concern[CA1, A1]. Specifically, we explained the transferability of MAPS across different disciplines and shared our thoughts on the issues related to formal language.
> >
> > As the rebuttal phase of ICLR is approaching its end, we would like to know if you have any further questions or suggestions about our paper. We would be more than happy to discuss them with you and make further improvements to our work.
> >
> > Best regards,
> >
> > The authors

---

### Official Review · Reviewer_1DpV · 2024-11-02

**Soundness:** 3
**Presentation:** 2
**Contribution:** 2
**Rating:** 6
**Confidence:** 3

**Summary:**

This paper presents MAPS (Multi-Modal Scientific Reasoning with Physics Perception and Simulation), a novel framework designed to improve the performance of Multi-Modal Large Language Models (MLLMs) in expert-level scientific reasoning tasks, specifically in physical sciences. MAPS enhances the comprehension and analytical processes by integrating a Physics Perception Model (PPM) with a simulator for interpreting complex physical diagrams and quantitative reasoning.

**Strengths:**

S1: The integration of perception and simulation for multi-modal scientific reasoning is well-conceived and leverages MLLM strengths while mitigating their weaknesses in handling complex diagrams.

S2: Results from circuit analysis problems highlight a notable increase in accuracy, showcasing MAPS' ability to outperform current state-of-the-art methods.

S3: The paper provides comprehensive explanations for data synthesis, PPM training, and the inference process, enhancing reproducibility.

**Weaknesses:**

W1: The framework is tested primarily on circuit analysis, which may not fully capture its adaptability across different physical sciences.

W2: The multi-step process involving diagram conversion, SL generation, and simulation may introduce cumulative errors, which could affect real-world applicability.

W3: The reliance on synthetic data poses a challenge for real-world accuracy, as unseen or complex diagrams might not align with generated examples.

**Questions:**

The paper demonstrates the effectiveness of MAPS using circuit analysis problems. Can the authors provide evidence or insights into how MAPS could be adapted for other physical sciences, such as mechanics or optics, which involve different types of diagrams and simulation languages?

---

> ### Author Response · Authors · 2024-11-21
> **Response to Reviewer 1DpV**
>
> We greatly appreciate the suggestions and questions from the reviewer.
>
> **[Q1]** **Adaptability across different physical sciences.**
>
> **[A1]** Please refer to our response to **[CQ2]** in common response to all reviewers.
>
> **[Q2] Concern about cumulative error.**
>
> **[A2]** As discussed in the Appendix, the errors within the MAPS framework primarily arise during the Chain-of-Simulation (CoS) phase, particularly due to errors in translating simulation language descriptions, and during the SAR phase, due to hallucinations in MLLM reasoning. Currently, the main bottleneck for MAPS lies in the accuracy of circuit diagram perception in the PPM phase.
>
> However, it is important to note that our goal is to answer expert-level questions in scientific domain, which involves complex diagram understanding and reasoning. Even humans are prone to mistakes in such challenging tasks. Approaches based on direct reasoning using MLLM also involve multi-step textual reasoning, which can result in cumulative errors. As mentioned in Section 2, the current capacity of MLLMs to describe physical scenarios and perform mathematical reasoning remains limited. The motivation behind MAPS is to alleviate these cumulative errors. Based on our experiment results and analysis, MAPS effectively mitigates hallucination issues in both the perception and mathematical reasoning stages, achieving significantly higher accuracy.
>
> In Section 7, we outlined the future directions for improving MAPS to further reduce cumulative errors and enhance its performance and feasibility in real-world scenarios. Specifically, we need to train more generalized and accurate PPMs to reduce errors during the CoS phase. Additionally, improvements in the foundational capabilities of MLLM will also reduce reasoning errors in MAPS.
>
> **[Q3]** **Generalization issue of PPM training on synthetic data.**
>
> **[A3]** Due to the lack of extensive datasets pairing schematic diagrams with corresponding simulation language descriptions, we resort to synthetic data for this process. As mentioned in Section 5.2, "Using VLM to implement PPM," the pre-trained VLM, when fine-tuned with synthetic data, demonstrates a encouraging level of generalization capability. Specifically: 1. Although the synthetic data only contains integer values, the PPM successfully translates floating-point numbers present in the test set diagrams; 2. The synthetic data only includes horizontal and vertical connections, yet the PPM successfully identifies components on slanted wires in the test set. However, overall, there is still significant room for improving the recognition accuracy of PPM in real-world scenarios.
>
> With regard, we propose two directions to enhance the generalization performance of PPM: 1. Collect more real-world data and manually annotate their simulation descriptions to train the PPM with a combination of synthetic and real data; 2. Optimize the distribution of synthetic data to better match real-world scenarios. These optimizations are part of our future work.
>
> Please let us know if the reviewer has any further questions, and we are more than happy to incorporate any additional suggestions that could further enhance our work and lead to a more favorable evaluation of our work.

---

> > ### Author Response · Authors · 2024-11-27
> >
> > Dear Reviewer 1DpV
> >
> > Thank you very much for your insightful feedback. During the rebuttal phase, We have provided detailed explanations for each of your concerns. Specifically, we have explained the following:
> >
> > - The insights of the generalization of MAPS across different disciplines [CA2].
> > - The advantages of MAPS in terms of cumulative error compared to existing methods. [A2]
> > - The necessity of using synthetic data in PPM training and the generalization performance exhibited by PPM. [A3]
> >
> > As the rebuttal phase of ICLR is approaching its end, we would like to know if you have any further questions or suggestions about our paper. We would be more than happy to discuss them with you and make further improvements to our work.
> >
> > Best regards,
> >
> > The authors

---

> > ### Comment · Reviewer_1DpV · 2024-12-02
> >
> > I find your rebuttal comprehensive and satisfactory. While challenges remain, your responses clearly articulate the current limitations and reasonable future plans for addressing them. Based on this, I will maintain my original score for the paper.

---

> > > ### Author Response · Authors · 2024-12-03
> > >
> > > We gratefully thank the reviewer for the insightful feedback and positive assessment of our work!

---

### Official Review · Reviewer_tWjp · 2024-11-04

**Soundness:** 3
**Presentation:** 3
**Contribution:** 2
**Rating:** 6
**Confidence:** 4

**Summary:**

This paper presents the framework of Multi-Modal Scientific ReAsoning with Physics Perception and Simulation (MAPS) based on an MLLM. It tries to decompose expert-level multi-modal reasoning task into physical diagram understanding via a Physical Perception Model (PPM) and reasoning with physical knowledge via a simulator.  The authors show that MAPS improves reasoning accuracy in the electronic circuit analysis domain. Towards that end, the authors craft rules to generate a large dataset of diverse circuit diagrams and their corresponding simulation language descriptions. This synthetic data is then used to fine-tune a pre-trained VLM, ultimately producing the PPM.

**Strengths:**

The paper makes several contributions. It creates a large dataset of diverse circuit diagrams.
It proposes to decompose expert-level multi-modal reasoning task into physical diagram understanding via a Physical Perception Model (PPM) and reasoning with physical knowledge via a simulator. (This is a good approach.)
The paper show that MAPS improves reasoning accuracy in the electronic circuit analysis domain.
The authors point out that all the information about a circuit may not be in the diagram and may be in the surrounding text and their approach addresses this issue.

**Weaknesses:**

It is not clear where errors happen. An error analysis of a sample of the evaluation data would be helpful. In particular,  please provide breakdown of errors by type (e.g., perception errors vs. reasoning errors), and give examples of common failure cases.

What are the different kinds of questions addressed in the paper? Do they need simulation or just a solver. (With simulation capability many more complex questions can be answered using multiple simulations.) Please provide a categorization of the question types in your dataset, along with examples of each category. In your current dataset what percentage of questions requires a single simulation, multiple simulations, and ones that can be solved with simpler methods.

What kind of training data is used with respect to the questions and Simulation language specification? More information on that would be helpful. In particular, please provide information such as the distribution of question types, and the complexity of circuits represented.

**Questions:**

See the weakness section.

The prompts in the Appendix mention "Chinese circuit problem".  What is meant by "Chinese circuit problem"? Is it a specific term? If not, perhaps you can revise its usage to avoid potential confusion.

---

> ### Author Response · Authors · 2024-11-21
> **Response to Reviewer tWjp [Part 1/2]**
>
> We appreciate the reviewer’s detailed questions and constructive suggestions. We will respond to each question in order as follows.
>
> **[Q1]** **Error Analysis**
>
> **[A1]** In Appendix C, we conducted an analysis of the error cases in MAPS and identified two fundamental types of errors: errors in Chain-of-Simulation (incorrect simulation language description conversion) and hallucinations in the Simulation-Aided Reasoning step. We provided two examples (Figure 19 and Figure 20) for each of these two error types. Notably, since the solver within the simulator is based on precise physical rules and numerical calculations, errors in the Chain-of-Simulation stage can only occur when generating the simulation language description.
>
> According to our analysis, the current major bottleneck of MAPS lies in the diagram perception component. Due to the widespread variations and complex structure of circuit diagrams, the PPM (trained on synthetic data) can still struggle to generalize to all diagrams in the test set. One of our next improvement directions is to further enhance the accuracy of PPM recognition.
>
> Compared to the initial submission, we have refined the analysis of error cases in Appendix C.3, with the revisions highlighted in red.
>
> **[Q2] Category of test questions. Do they require single simulation or multiple simulations?**
>
> **[A2]** As mentioned in Section 4.1, our test set `SimpleCircuitEval` is randomly selected from 79 linear pure resistive circuit (LPRC) analysis problems in the textbook, corresponding to four chapters. And the content and characteristics of these chapters are summarized as follows:
>
> | Chapter | Title | # Components (Avg.) | Characteristics |
> | --- | --- | --- | --- |
> | 1 | Circuit elements and circuit laws | 4.7 | Basic circuits. No controlled sources. |
> | 2 | Analysis method of simple resistance circuit | 6.8 | Resistance circuits. Not directly simulable, requires calculation of equivalent resistance. |
> | 3 | General method of analysis of linear resistance circuits | 7.6 | LPRC circuits. Normal topologies. |
> | 4 | Some theorems of electric circuits | 6.2 | LPRC circuits. Complex topologies. |
>
> We have included this table in Appendix C.1. Furthermore, for each of the four chapters, we provided example problems, as shown in Figure 15.
>
> The MAPS method proposed in this work is designed to perform a single simulation, as described in the Chain-of-Simulation process in Section 3.1. Regarding whether multiple simulations can solve more complex problems, we provide an intuitive discussion in Section 7. Currently, MAPS functions as a static workflow that obtains results through a single simulation. However, if we enable the LLM to interact with the simulator, leveraging feedback from simulation results and conducting multiple simulations, it may be possible to solve more complex problems (e.g., setting circuit component parameters as needed). This is a potential direction for future improvement of this work. We highly appreciate the reviewer’s very constructive question.

---

> > ### Author Response · Authors · 2024-11-21
> > **title: Response to Reviewer tWjp [Part 2/2]**
> >
> > **[Q3]** Training data of PPM. (converting diagrams to simulation language description)
> >
> > **[A3]** As mentioned in Sections 3.2 and 4.1 of our paper, we synthesized the training data for the Physics Perception Model (PPM) through our proposed data synthesis methods. Specifically, we automatically generated a large number of diverse circuit diagrams and their corresponding simulation language descriptions, taking into account the specific characteristics of the curcuits analysis discipline.
> >
> > In this work, to address circuit analysis problems in the LPRC scenario, we synthesized a dataset of 20k circuit diagrams paired with simulation language descriptions, named `ppm-syn-lprc`, which was used for PPM training. Details of the implementation can be found in Section 4.1 and Appendix B.1. Furthermore, in Appendix B.1, we analyzed the distribution of the synthesized data to ensure its diversity. Table 5 in Appendix B.1 provides the complexity metrics of the synthesized circuit diagrams, including the number of nodes, components, and measurements. Figure 13 in Appendix B.1 reports three specific synthesized circuit diagrams from the `ppm-syn-lprc` dataset.
> >
> > **[Q4**] Confusion about term "Chinese circuit prompt" in prompt templates.
> >
> > **[A4]** We sincerely apologize for the confusion caused by the wording used in our appendix. Please refer to [CA1] in Common Response to All Reviews comment for details of our revision.
> >
> > If the reviewer have any further questions or need additional clarification, please don't hesitate to reach out. We're eager to engage in further discussion and address any inquiries you may have!

---

> > > ### Author Response · Authors · 2024-11-27
> > >
> > > Dear Reviewer tWjp:
> > >
> > > Thank you very much for your thoughtful suggestions. Your feedback has significantly enriched the content of our article. Based on your advice, we have made the following improvements:
> > >
> > > - We conducted a more comprehensive analysis of the error types in MAPS. [A1]
> > > - We provided a more detailed introduction to our evaluation dataset. [A2]
> > > - We removed the potentially confusing terms "Chinese Circuit Problem" and "Chinese Question" in both prompt templates and case studies in the Appendix C. [CA1]
> > >
> > > As the rebuttal phase of ICLR is approaching its end, we would like to know if you have any further questions or suggestions about our paper. We would be more than happy to discuss them with you and make further improvements to our work.
> > >
> > > Best regards,
> > >
> > > The authors

---

> > > > ### Comment · Reviewer_tWjp · 2024-11-27
> > > > **Increasing my evaluation score**
> > > >
> > > > After reading your response I am increasing my evaluation score.

---

> > > > > ### Author Response · Authors · 2024-11-28
> > > > >
> > > > > We sincerely thank the reviewer again for the timely and constructive suggestions and appreciation of our work!

---

### Official Review · Reviewer_xCVY · 2024-11-04

**Soundness:** 4
**Presentation:** 4
**Contribution:** 3
**Rating:** 8
**Confidence:** 4

**Summary:**

This paper proposes an innovative framework named MAPS, which relies on multimodal large language models (MLLMs). During the inference phase, it integrates the simulated language descriptions of input diagrams generated by PPM and data obtained through the Chain-of-Simulation process, aiming to address complex structural and quantitative analysis issues in the field of physics. Through empirical studies on university-level circuit analysis problems, the authors demonstrate the significant effect of MAPS in enhancing the inference accuracy of MLLMs, outperforming all existing models. These results reveal the potential of MAPS in strengthening the multimodal scientific reasoning capabilities of MLLMs, providing a promising direction for the development of this field.

**Strengths:**

The advantages of this paper are mainly reflected in the following aspects:
- This paper proposes an innovative process framework that can combine physical perception models (PPMs) with simulator outcomes to infer answers to physical problems. The framework integrates the understanding of physical diagrams with the reasoning of physical knowledge, and its effectiveness has been validated through experiments;
- The paper designs and introduces a synthetic dataset named ppm-syn-lprc, which is used for fine-tuning the visual-language model PPM. The generation process of the synthetic dataset is elaborated in detail, and the scalability of this method is demonstrated. For instance, this approach can be applied to other fields such as mechanical systems, providing clear directions and ideas for improving the model's reasoning capabilities in other areas of physics;
- The structure of the paper is well-organized and logical, making it easy for readers to understand and follow.

**Weaknesses:**

Writing:
- There are some typos in the article, and it is recommended that the author carefully proofread and corrected them to enhance the professionalism and readability of the paper.

Experimental Design:
- Generalization Issues: The results of this study have only been validated on the GPT-4V model, which may not be sufficient to demonstrate the applicability of the framework to other model architectures. It is suggested that the authors extend the evaluation of the framework to different model architectures to strengthen the generalizability and credibility of the experimental conclusions.
- Evaluation of Synthetic Data Effectiveness: The paper mentions using CogVLM-17B as the base model for PPM and fine-tuning it on the synthetic dataset. To clarify whether the improvement in the model’s ability to understand physical diagrams is due to the inherent capabilities of the base model or the introduction of synthetic data, the authors are advised to add comparative experiments before and after fine-tuning. This will help verify the effectiveness of synthetic data and reinforce the persuasiveness of the research findings.

**Questions:**

Incorrect spelling and expression:
- Line 243 - diagram diagram -> diagram
- Appendix A.2, Figure 5, Row2 - Chnese -> Chinese

Some Confusions:
The research idea proposed in this paper is highly innovative, but during my reading, I encountered several questions that I hope the authors can provide answers to:
- Table 1 presents the conversion effectiveness of PPM on the synthetic dataset ppm-syn-lprc-test and the real-world dataset SimpleCircuitEval. It is noted that the recognition accuracy of Label-Type diagrams is lower than that of Numerical-Type diagrams, which seems contrary to common belief since label-type diagrams are generally considered easier to identify than numerical-type diagrams. What is the reason for this difference?
- The test dataset SimpleCircuitEval comprises only 79 questions. Is this sample size sufficient to support a convincing evaluation result? Does it adequately represent the universality of LPRC question types?

---

> ### Author Response · Authors · 2024-11-21
> **Resposne to Reviewer xCVY [Part 1/2]**
>
> We sincerely appreciate Reviewer xCVY's comments, which are instrumental for further improving our paper. We first thank the reviewer for identifying the typo. We have made the careful corrections. Please refer to the comment "Common Response to Reviewers", response [CA1], for details.
>
> Below, we provide detailed responses to each of the reviewer's questions and outline the corresponding revisions made to our manuscript.
>
> **[Q1] Generalization issues across different model architectures.**
>
> **[A1]** We initially validated the MAPS framework on GPT-4V as it was and is still one of the most advanced models for multimodal understanding, allowing us to observe more pronounced effects. We appreciate the reviewer's suggestion regarding the importance of MAPS performance across different architectures. Accordingly, we have tested the MAPS framework on two structurally distinct MLLMs: Claude-3.5 and GLM-4V. The results are shown in the table below:
>
> |  | Chap1 | Chap2 | Chap3 | Chap4 | Avg. |
> | --- | --- | --- | --- | --- | --- |
> | GPT-4V | 16.0 | 4.17 | 5.26 | 9.09 | 7.6 |
> | GPT-4V + MAPS | 52.0 | 7.14 | 36.8 | 45.5 | 32.9(x4.3) |
> | Claude-3.5 | 16.0 | 4.17 | 0.0 | 0.0 | 6.33 |
> | Claude-3.5 + MAPS | 44.0 | 12.5 | 21.1 | 36.4 | 25.3(x4.0) |
> | GLM-4V | 8.0 | 4.17 | 5.26 | 0.0 | 6.33 |
> | GLM-4V + MAPS | 32.0 | 0.0 | 15.8 | 36,4 | 19.0(x3.0) |
>
> As illustrated, MAPS improved the accuracy of circuit analysis problem-solving by more than threefold across different MLLM architectures (GPT-4V, Claude-3.5, GLM-4V). In our experiments, we maintained the same PPM and prompt templates to ensure fairness. The results indicate that MAPS possesses promising cross-model transferability. However, variations in multi-modal content comprehension capabilities across different base MLLMs can impact MAPS performance. For instance, GLM-4V's comprehension ability is relatively weaker compared to GPT-4V and Claude-3.5. When dealing with MAPS simulation and solution prompts, GLM-4V often fails to comprehend them correctly and outputs "unable to compute," thus unable to perform further inference and lowering the accuracy.
>
> We have incorporated the results from different MLLMs into our main experimental table (Section 4.3, Table 2). We would like to thank Reviewer xCVY for this valuable suggestion. It has clearly strengthened our experiment results and further substantiated the effectiveness of MAPS.

---

> > ### Author Response · Authors · 2024-11-21
> > **Resposne to Reviewer xCVY [Part 2/2]**
> >
> > **[Q2] Evaluation of Synthetic Data Effectiveness.**
> >
> > **[A2]** This is an excellent question! However, our observations indicate that the accuracy of existing visual language models (VLMs) in converting physical diagrams into simulation language descriptions is nearly zero. This is potentially because this task is substantially outside the distribution of the VLM’s pre-training corpus, as we have mentioned in Section 3.2 and Section 5.1 Question 1. Although our manuscript primarily referred to GPT-4V, preliminary experiments have shown that CogVLM-17B (base model of PPM) is similarly incapable of directly translating diagrams into simulation language descriptions, much like GPT-4V’s performance. We have included these findings in Section 4.1 (Paragraph: PPM Training) to strengthen our paper. Please refer to our updated manuscript for more details.
> >
> > **[Q3] Reason behind the lower recognition accuracy of Label-Type diagrams compared to Numerical-Type diagrams.**
> >
> > **[A3]** In this work, we only compared Label-Type diagrams and Numerical-Type diagrams using the $ACC_{CQ}$ metric. The $ACC_{CQ}$ metric primarily measures the quantity of components, without accounting for the correctness of the recognized component labels or numerical values. This metric therefore only provides a rough measure of circuit recognition accuracy.  In fact, based on our observations from `SimpleCircuitEval`, there is no significant difference in recognition accuracy between Label-Type diagrams and Numerical-Type diagrams.
> >
> > **[Q4] Concern about the size of the evaluation set.**
> >
> > **[A4]** To evaluate the performance of MAPS on Linear Pure Resistive Circuits (LPRC), we selected questions involving LPRC from the first four chapters of a college-level circuit analysis textbook. To ensure question diversity and coverage, we consulted domain experts in the related field to remove redundant questions, resulting in a final set of 79 questions. We should note that in many existing LLM benchmarks, around 80 representative test cases are not considered a small scale, especially in very domain-specific settings; for example, the well-known test sets MT-bench [R1] and HumanEval [R2] contain 80 and 164 questions, respectively.
> >
> > In the updated version of our paper, we have included the characteristics and examples of our test cases from all four chapters. Please refer to Appendix C.1 for more details.
> >
> > References:
> >
> > [R1] Zheng, Lianmin, et al. "Judging llm-as-a-judge with mt-bench and chatbot arena." *Advances in Neural Information Processing Systems* 36 (2023): 46595-46623.
> >
> > [R2] Chen, Mark, et al. "Evaluating large language models trained on code." *arXiv preprint arXiv:2107.03374* (2021).

---

> > > ### Comment · Reviewer_xCVY · 2024-11-25
> > >
> > > Thank you to the authors for their detailed response. I am pleased to see that all my concerns have been thoroughly addressed and supported by new experimental results. Based on this, I will reassess and update my rating for this paper. Based on this, I will update my rating for this paper.

---

> ### Author Response · Authors · 2024-11-25
>
> Thanks for the reviewer’s reply and we are delighted to hear that the reviewer's questions have been satisfactorily addressed! Please let us know if the reviewer has any further questions, and we are more than happy to incorporate any additional suggestions that could further improve our work.

---

> ### Author Response · Authors · 2024-11-25
>
> Dear Reviewer xCVY,
>
> We are pleased the our responses have addressed your concerns, and you also indicated that you would update the score. However, we have noticed that the score has not yet been updated on the OpenReview platform. We would like to know if there is anything else you would like us to do before you update the score, given the author response period is fast coming to its end.
>
> Best regards,
>
> The authors

---

> > ### Comment · Reviewer_xCVY · 2024-11-30
> >
> > I have updated my rating.

---

> > > ### Author Response · Authors · 2024-12-02
> > >
> > > We sincerely appreciate your thoughtful and constructive feedback, as well as your positive assessment of our work!

---

### Author Response · Authors · 2024-11-21
**Common Response to All Reviewers**

We sincerely thank all reviewers for their constructive feedback and detailed comments. We are encouraged to find that our reviewers appreciate the novelty of our proposed idea (Reviewers xCVY, b9qd) with remarkable performance in real-world scenarios (Reviewers xCVY, tWjp, 1DpV, b9qd), and clear presentation (Reviewers b9qd, xCVY, 1DpV).

It is worth noting that there are some shared questions regarding the writing of our paper and domain transferability across different physical disciplines. Next, we provide our clarifications to these common questions (CQs) in the general response, followed by individual responses to each reviewer's specific questions.

**[CQ1] Typos & Confusing wordings**

**[CA1]** Despite our careful revisions before the submission, there were still a few typos and confusing wordings remained in our paper. We're grateful for Reviewer xCVY and tWjp to bring these to our attention. We've made careful corrections in our updated manuscript. Specifically, we've fixed the typo "diagram diagram" and removed the confusing term "Chinese Circuit Problem" from our prompt (Appendix A.2, Figures 5, 6, 7, and 8). We sincerely appreciate the reviewers' suggestions, which have significantly enhanced the clarity and quality of our paper.

**[CQ2] Adaptability across different science domains**

**[CA2]** This is a very important question. As mentioned in Section 7, because the MAPS framework requires a robust understanding model of physical schematics, i.e. PPM, which in turn counts on a sophisticated data synthesis process incorporating domain-specific knowledge for model training, our current work has conducted experiments only in the circuit analysis domain. Technically, the MAPS framework can be naturally extended to many other physical disciplines: as long as we can describe the schematic's underlying physical dynamics using formal simulation language and leverage corresponding simulators to carry out the corresponding physical system, we can apply the proposed solutions.

The key to applying the MAPS framework across different disciplines lies in effective engineering implementations. Based on our circuit analysis experience, we would like to provide the following insights for extending MAPS to other disciplines:

1. We need to design the synthesis of PPM training data tailored to the characteristics of schematics in different disciplines to train the corresponding PPM. In Section 5.2, "Philosophy of PPM Construction," we outlined the fundamental principles. First, we need to  establish the components and layouts specific to each discipline. For example, in a statics system, components might include blocks, rods, ropes, hinges, and spheres, with positional relations such as connections and tangency. To generate a <schematic, simulation language>  pair, we can exhaust the combination of the components and their layouts, apply physical rules and simulation language syntax to generate the corresponding simulation language descriptions and diagram generation instructions (such as latex), and subsequently create the corresponding image.
2. We need to configure different physical simulators tailored to different scenarios. For instance, for a statics system, we might use the ANSYS simulator to model the scenario, while for an optical system, we could use Zemax OpticStudio for simulation. This step primarily involves designing a simulation result format that is easily understandable and inferable by LLMs. In this paper, we represent simulation results within the circuit analysis domain using JSON dictionaries to facilitate model comprehension.

---

> ### Comment · Reviewer_tWjp · 2024-11-25
>
> The revised version still has the phrases "Chinese Circuit Problem" (see for example, Figure 20) and that example also has a phrase "Chinese question ...".

---

> > ### Author Response · Authors · 2024-11-26
> >
> > We sincerely apologize for our oversight, which resulted in the confusing terminology remaining in the case study images. We highly appreciate Reviewer tWjp's review, and we have removed the confusing terms "Chinese circuit problem" and "Chinese question" from all case studies (Figure 16, 17, 18, 19, 20). Thank you again for your careful examination.

---

### Author Response · Authors · 2024-11-25
**A Gentle Reminder to the Reviewers**

Dear reviewers,

We hope this message finds you well. We understand that the author reviewer discussion is a critical component of the ICLR review process, and we would like to remind you that the rebuttal period is scheduled to **conclude on November 26th**. Given there are only 2 days left before the deadline, we would like to call for your attention to our provided responses.

Thank you again for your valuable time and thoughtful comments. We have provided comprehensive responses and additional results in light of your feedback. As we are nearing the end of the discussion stage, we would appreciate it if you could review our responses and update your scores if your concerns have been adequately addressed. We remain open to further discussion should there be any issues that you feel have not been fully resolved. Thank you for your consideration.

Best regards,

The Authors

---

### Meta-Review · Area_Chair_Suic · 2024-12-05

**Metareview:**

**(a) Summarize the scientific claims and findings of the paper.**

The paper presents the Multi-Modal Scientific Reasoning with Physics Perception and Simulation (MAPS) framework, designed to enhance the reasoning capabilities of Multi-Modal Large Language Models (MLLMs) in the physical sciences, specifically for circuit analysis. MAPS integrates two components: a Physics Perception Model (PPM) trained on synthetic data to interpret physical diagrams and a simulator to reason about physical knowledge via a Chain-of-Simulation process. The approach demonstrates significant improvements over existing models in reasoning tasks for circuit analysis, validating its effectiveness in expert-level multi-modal reasoning tasks.

**(b) Strengths**

1. **Novel Framework**: MAPS combines physical perception with simulation-based reasoning, addressing limitations in MLLMs for domain-specific tasks.
2. **Performance Gains**: Significant improvements in circuit analysis problems showcase the potential of MAPS in scientific reasoning tasks.
3. **Synthetic Data Generation**: The paper details a scalable method for generating synthetic datasets to fine-tune the PPM.
4. **Cross-Disciplinary Potential**: While demonstrated in circuit analysis, MAPS is conceptually extensible to other physical sciences with formalized rules and simulation tools.
5. **Clear Presentation**: The paper is well-organized and provides comprehensive explanations of the framework and its components.

**(c) Weaknesses**

1. **Limited Domain Coverage**: The evaluation focuses solely on circuit analysis, raising questions about generalizability to other disciplines.
2. **Error Analysis**: While errors are identified, the breakdown and mitigation strategies could be more detailed, particularly regarding perception and reasoning failures.
3. **Synthetic Data Dependence**: The reliance on synthetic data limits real-world applicability, as unseen or complex diagrams might diverge from the generated examples.
4. **Scalability Concerns**: Extending MAPS to disciplines without formal language descriptions or widely used simulators remains a challenge.

**(d) Reasons for the decision.**

MAPS is a novel and promising method that significantly improves multi-modal reasoning in circuit analysis and has the potential for broader scientific applications. While there are limitations in domain coverage and scalability, the strengths in innovation, methodology, and performance gains justify acceptance with a spotlight recommendation, and inspire the community to improve visual reasoning capabilities with a similar approach.

**Additional Comments On Reviewer Discussion:**

The reviewers raised key concerns about generalizability, scalability, and error analysis. The authors addressed these points effectively:

1. **Generalization Across Disciplines**: The authors elaborated on the adaptability of MAPS to other physical sciences, providing insights into the extension process.
2. **Error Analysis**: An improved breakdown of errors, focusing on the Chain-of-Simulation and reasoning steps, was included in the revised manuscript.
3. **Synthetic Data Dependence**: The authors outlined future directions, such as combining synthetic and real-world data, to enhance the generalizability of the PPM.

The authors' responses and additional experiments bolstered confidence in the paper's contributions, though some concerns, such as scalability to disciplines without formal languages, remain open.

---

### Decision · Program_Chairs · 2025-01-22

Accept (Poster)